# Beyond the microcirculation: sequestration of infected red blood cells and reduced flow in large draining veins in experimental cerebral malaria

A. M. Oelschlegel[1,2,10], R. Bhattacharjee[1,3,10], P. Wenk[1,10], K. Harit[3], H-J Rothkötter[4], S. P. Koch[5,6,7], P. Boehm-Sturm [5,6,7], K. Matuschewski [8], E. Budinger[1,9], D. Schlüter[3], J. Goldschmidt [1,9] ✉ & G. Nishanth [3] ✉

Sequestration of infected red blood cells (iRBCs) in the microcirculation is a hallmark of cerebral malaria (CM) in *post-mortem* human brains. It remains controversial how this might be linked to the different disease manifestations, in particular brain swelling leading to brain herniation and death. The main hypotheses focus on iRBC-triggered inflammation and mechanical obstruction of blood flow. Here, we test these hypotheses using murine models of experimental CM (ECM), SPECT-imaging of radiolabeled iRBCs and cerebral perfusion, MR-angiography, q-PCR, and immunohistochemistry. We show that iRBC accumulation and reduced flow precede inflammation. Unexpectedly, we find that iRBCs accumulate not only in the microcirculation but also in large draining veins and sinuses, particularly at the rostral confluence. We identify two parallel venous streams from the superior sagittal sinus that open into the rostral rhinal veins and are partially connected to infected skull bone marrow. The flow in these vessels is reduced early, and the spatial patterns of pathology correspond to venous drainage territories. Our data suggest that venous efflux reductions downstream of the microcirculation are causally linked to ECM pathology, and that the different spatiotemporal patterns of edema development in mice and humans could be related to anatomical differences in venous anatomy.

Cerebral malaria (CM) is a severe neurological complication of human malaria mostly caused by the parasite *Plasmodium falciparum* (*Pf*). It is the most common non-traumatic encephalopathy in the world with a high mortality rate. Globally, an estimated number of 608,000 patients died of malaria in 2022, while 96% of the malaria-related deaths occurred in sub-Saharan Africa, primarily in infants and children below the age of 5 years (WHO, 2023[1]).

Patients with acute infection can present with a diffuse CM encephalopathy, a rapid progressive coma, and seizures without return to consciousness[2,3]. Magnetic resonance imaging (MRI) findings in children in sub-Saharan Africa indicate that increased brain volume and brain swelling are hallmarks of the disease, and patients die from brain herniation[4,5]. MRI measures of brain edema strongly correlate with the outcome in pediatric CM[6].

The key pathological findings in *post-mortem* human brains are intravascular sequestration of and congestion with infected red blood cells (iRBCs)[7–11]. In contrast to other parasitic diseases of the CNS, *Pf*-infected red blood cells in CM remain primarily intravascular[2,3,12,13].

However, in murine experimental CM (ECM) some iRBCs have also been observed perivascularly in the parenchyma[14]. The cascade of events linking infection and brain swelling has remained elusive and is much debated[4,15]. Theories focus either on a cytokine storm triggering inflammation and affecting vascular integrity or mechanical obstructions of cerebral blood flow (CBF) due to iRBC sequestration in the microcirculation[2,3,12,13]. The former is primarily based on ECM models, while the latter originates from *post-mortem* histology of CM patients. While the relevance of ECM to CM continues to be viewed differently[16,17], there is growing consensus that studies with experimental models of CM are central to better define pathogenesis and identify early signatures that distinguish disease progression from parasite propagation[14,18]. One central controversy is to which extent iRBC sequestration in cerebral microcapillaries, a distinctive hallmark of CM, also contributes to the disease in ECM[12,19–24]. Therefore, insights into the temporal development of disease events in combination with their regional distribution might provide a better understanding of the pathophysiological processes leading to CM.

As the spatial patterns of brain swelling differ in different subgroups of patients[4], it could be highly informative to correlate at, or close to, the onset of cerebral edema, the spatial patterns of iRBC sequestration with spatial patterns of edema development and potential perfusion deficits, which are to be expected upon mechanical obstruction. However, such studies are hardly feasible in humans, especially in the most affected countries with limited access to the imaging technologies needed.

Herein, we address these questions in murine ECM focusing on *Plasmodium berghei* ANKA (*Pb*A) infected C57BL/6 wild-type (wt) mice. We extend parts of our study to murine malaria models that do not develop ECM or brain pathology. After studying brain iRBC accumulation in infected mice using quantitative real-time-PCR (q-PCR), we use single-photon emission computed tomography (SPECT) for imaging in vivo the distribution of $^{99m}$Technetium ($^{99m}$Tc)-labeled iRBCs in the brain at day 5 post-infection (p.i.), an early stage of the disease when first clinical symptoms start appearing. We combine this approach with T2-weighted (T2w) MRI for edema monitoring as well as SPECT-imaging of CBF and MR-angiography. These imaging approaches allow us to detect potential sequelae of iRBC sequestration on cerebral perfusion at different spatial levels, *i.e.*, capillary flow and large draining veins. We also relate these data to histological data at different time points after infection and to the expression of inflammation markers, as studied by q-PCR and immunohistochemistry, with and without anti-malarial treatment.

## Results

### Early-stage accumulation of *Pb*A-infected red blood cells in large cerebral draining veins in C57BL/6 wt mice

We initiated our study by q-PCR analysis and quantified the *Pb*A-specific gene cytochrome B in the brains of infected C57BL/6 wt mice at days 5 and 7 p.i. and, as control, in uninfected mice. Abundance of parasite mRNA was quantified in three different cerebral regions, olfactory bulb, cerebral cortex, and brainstem (Fig. 1A). Interestingly, *Pb*A cDNA could be detected as early as day 5 p.i., prior the onset of noticeable disease symptoms (Fig. S1). As the disease progressed until day 7 p.i., the parasite load increased in all three regions of the brain. Of note, the parasite burden remained significantly higher in the olfactory bulb compared to cerebral cortex and brainstem (Fig. 1A).

In order to study the spatial distribution of iRBCs at an early stage in more detail, we labeled *Pb*A-infected RBCs with the gamma-emitter $^{99m}$Tc and studied their biodistribution in *Pb*A-infected C57BL/6 wt mice in vivo using SPECT. 1 h after injection of the labeled iRBCs, we imaged the $^{99m}$Tc-distributions in the heads of uninfected and infected mice at day 5 p.i. An accompanying whole-body scan was used to precisely determine standardized uptake values (SUVs). SUVs represent the ratio of the $^{99m}$Tc-concentration in a voxel and the whole-body

$^{99m}$Tc-concentration. Peak differences in $^{99m}$Tc-content between infected mice and controls were found in areas corresponding to large cerebral draining veins or sinuses, in particular the rostral rhinal veins, the superior sagittal sinus, the straight sinus, and the transverse sinuses (Fig. 1B–F). The distribution of the differences within the superior sagittal sinus was inhomogeneous, strongly favoring the T-shaped junctions at the rostral and caudal confluences of the sinuses (Fig. 1B–F). An additional lower-intensity peak was found in the middle of the sinus (Fig. 1B–D). Literature data suggest that this peak could correspond to an entry site of larger draining veins[25].

We then imaged, as further controls, the iRBC distribution in *Pb*A-infected perforin-deficient mice (C57BL/6 *prf*$^{-/-}$) (Fig. 1G–I) and BALB/c mice (Fig. 1J, K). In both C57BL/6 *prf*$^{-/-}$ and BALB/c mice *Pb*A replicates in erythrocytes and causes anemia, but not ECM[26,27]. The spatial distribution of the iRBCs in C57BL/6 *prf*$^{-/-}$ mice closely mimicked that of the C57BL/6 wt mice (Fig. 1G–I), but the ratio of iRBCs in infected compared to non-infected mice was lower in C57BL/6 *prf*$^{-/-}$ mice (Fig. 1L). No differences were found between infected and uninfected BALB/c mice (Fig. 1J–L). The course of parasitemia was comparable between the infected C57Bl/6 wt mice and BALB/c mice (Fig. S2B), indicating that the genetic background of the host determines the outcome of the vascular pathogenesis in ECM.

Together, our data on the distribution of $^{99m}$Tc-labeled iRBCs identified accumulation in large cerebral draining veins at an early time point prior to clinical signs of progression to ECM. In infection models that progress to malaria-related anemia instead of ECM this accumulation was either less pronounced (C57BL/6 *prf*$^{-/-}$ mice) or absent (BALB/c mice).

### Reduced flow in large draining cerebral veins in *Pb*A-infected mice

Our iRBC-SPECT findings suggested that venous efflux in large draining veins might be impaired already at an early stage of disease onset indicative of a trajectory towards ECM. We tested this hypothesis using time of flight MR-angiography (TOF-MRA) with sequences for imaging blood flow in caudo-rostral direction in C57BL/6 wt, C57BL/6 *prf*$^{-/-}$ and BALB/c mice (Fig. 2). We used a novel automated workflow atlas registration described previously by Koch and colleagues[28] for aligning all MR-angiographies in the same anatomical reference frame. We adopted this workflow to the present ECM study, which greatly facilitated group-level analyses at single-vessel levels (Fig. 2).

For quantitative analysis, we measured the signal intensities in a volume-of-interest (VOI) segmented from the superior sagittal sinus (sss) and the rostral rhinal vein (rrv) of uninfected control mice. In this VOI, the mean signal intensity dropped by 22% in infected C57BL/6 wt mice at day 5 p.i. compared to controls ($p < 0.01$), and further dropped by 39% at day 7 p.i., as compared to controls ($p < 0.001$) (Fig. 2). Peak intensities inside the sss and rrv dropped by more than 70% at day 7 p.i. compared to the uninfected controls. The variance at day 7 p.i. was the highest of all groups measured, and in some individuals the flow was barely detectable.

In our anatomical T2w images (Figs. 2 and 3) we found prominent signal increases indicating severe edema at day 7 p.i. in the same structures as described previously[29], in particular in white matter fiber tracts, such as the corpus callosum (Figs. 2 and 3). At day 5 p.i. we were unable to unambiguously identify or exclude signs of edema. In line with a previous study[30], we found no evidence for edema in infected C57BL/6 *prf*$^{-/-}$ mice. Although mean signal intensity in the sss-rrv VOI decreased slightly, the decreases did not pass thresholds of significance (Fig. 2). In infected BALB/c mice, we detected a slight, albeit non-significant, increase in mean signal intensity (Fig. 2). This might simply reflect noise but could also be related to blood flow alterations as a consequence of the systemic infection.

In all groups of mice, we noted that the sss gives of a branch that runs above this sinus into the rrv (Fig. 3 d0 Bregma +3.0 and +0.8;

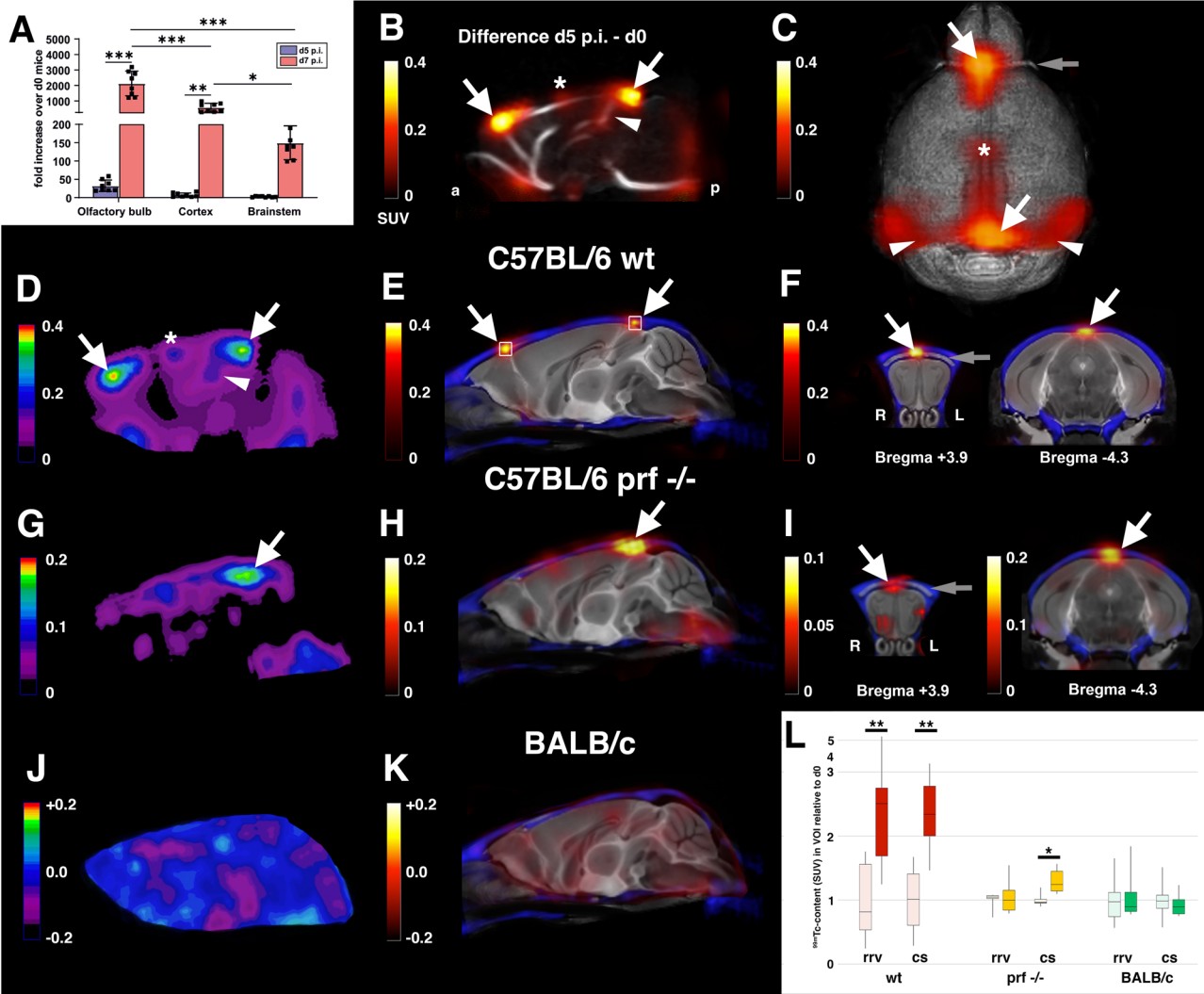

**Fig. 1 | *Pb*A-infected red blood cells accumulate at an early stage in large draining veins of C57BL/6 wild-type and C57BL/6 *prf*<sup>-/-</sup> mice, but not BALB/c mice. A** Quantitative real-time-PCR analysis of *Pb*A cytochrome B mRNA expression in selected regions of the brains of infected C57BL/6 wt mice at d5 p.i. and d7 p.i. Data show the increase of the respective mRNA expression of *Pb*A-infected over uninfected mice (d0) of the same mouse strain. *n* = 4 independent animals with 2 replicates each for all groups. Data are presented as mean values ± SD. **B–F** Spatial distribution of the differences in $^{99m}$Tc-labeled iRBC contents between mice at day 5 p.i. and day 0 in C57BL/6 wt mice, **G–I** C57BL/6 *prf*<sup>-/-</sup> mice and **J, K** BALB/c mice. All color scales are in units of SUV. All SPECT images are differences of group-mean d5 p.i. minus d0 data. **B** Overlay of a midsagittal section of the SPECT-difference image on a group-mean MR-angiography from d0 mice. **C** Top view on a volume rendered difference SPECT overlaid on the reference MR. **D, G, J** Midsagittal sections of the difference SPECT in the different groups.

**E, F, H, I, K** For precise localization of the peak regions, the SPECT-difference data sets were overlaid on an anatomical reference MR. The MR dataset is from an ex vivo measurement[76] and includes a signal-intense rostral rhinal vein (gray arrows in **C, F,** and **I**). Note the precision of the alignment of the group-mean CTs (blue) and the MR (**F, I**). Peak differences in labeled iRBCs are found at the rostral (arrows pointing right downward) and the caudal confluences of the sinuses (arrows pointing left downward). Note also increased accumulation in midsagittal sinus (asterisk in **B–D**), straight sinus (arrowhead in **B, D**), and transverse sinus (arrowheads in **C**). **L** Boxplot diagram (min, max. median, 1st and 3rd quartil) of accumulation of $^{99m}$Tc-labeled iRBCs in rrv and confluence of the sinuses (cs) VOIs in d5 p.i. relative to d0 mice (C57BL/6 wt d0 *n* = 7, d5 *n* = 8; C57BL/6 *prf*<sup>-/-</sup> and BALB/c *n* = 6, 9 weeks old female mice in all groups). The VOIs are shown in the sagittal section in (**E**) (white squares). **p* < 0.05, ***p* < 0.01, ****p* < 0.001, two-way ANOVA in (**A**), two-tailed unpaired heteroscedastic *t*-test in (**L**).

---

Fig. 2, BALB/c mice d0). The mean signal intensity in all groups was higher in this upper branch as compared to the lower (Fig. 2).

We conclude from the TOF-MRA data that in *Pb*A-infected C57BL/6 wt mice a reduction in venous efflux precedes edema formation and progresses rapidly, corresponding to the severity of the disease, as quantified by clinical scores (Fig. S1).

**Histology confirms the presence of iRBCs in large draining veins of *Pb*A-infected C57BL/6 wt mice**
We then looked for vascular pathological alterations in the large draining veins and studied *Pb*A-infected C57BL/6 mice histologically (Fig. 4). We reasoned that in order to not miss pathological

intravascular phenomena, for instance rosetting, we should omit transcardial perfusion. Our analysis was based on immersion-fixed decalcified material.

At day 7 p.i., when the mice showed typical symptoms of ECM with a mean RMCBS score of 8, we found iRBCs with dark black-brown deposits with diameters of several micrometers most likely representing hemozoin in the large draining veins of *Pb*A-infected C57BL/6 wt mice (Fig. 4A–J). IRBCs were also present in the large draining veins of *Pb*A-infected C57BL/6 *prf*<sup>-/-</sup> mice, but we were unable to find the large-diameter deposits that characterized the C57BL/6 wt iRBCs (Fig. 4K–U). To further clarify whether the presence of these deposits might be related to or correlate with the development of the ECM

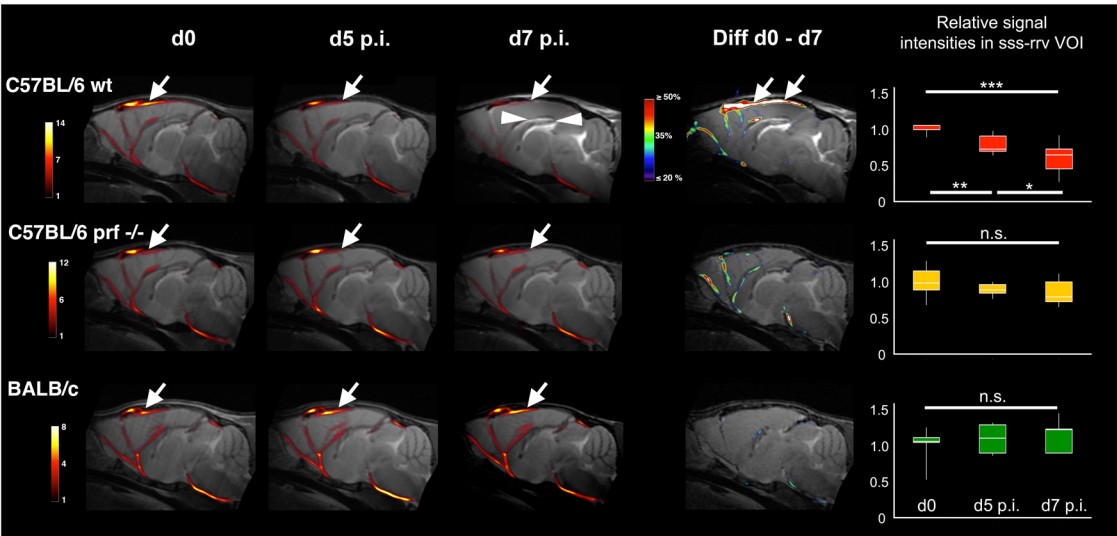

**Fig. 2 | Magnetic resonance angiography uncovers reduced flow in large draining veins only in *Pb*A-infected mice on trajectory to ECM.** Midsagittal sections of group-mean MR-angiographies from mice at d0, d5 p.i. and d7 p.i. and differences between d0 and d7 overlaid on T2w MR-images from single individuals of the according groups in C57BL/6 wt (upper row), C57BL/6 *prf*$^{-/-}$ (middle row) and BALB/c mice (lower row). Color scales are in signal intensities of the TOFs and in percentage difference in Diff d0-d7. Arrows point to the sss. Boxplot diagrams of changes in signal intensities relative to d0 in a volume-of-interest (VOI) comprising sss and rrv are displayed on the right. In C57BL/6 wt mice signal intensities in the sss and rrv (arrows) drop significantly already at day 5 p.i. and are severely reduced at day 7 p.i. Note that edema formation, including in the *corpus callosum* (arrowheads), is unequivocally detectable on day 7 in *Pb*A-infected C57BL/6 wt mice. In C57BL/6 *prf*$^{-/-}$ mice the mean signal intensity drops at d5 p.i. and d7 p.i., but does not pass significance thresholds. (C57BL/6 wt d0 $n = 5$, d5 $n = 10$, d7 $n = 10$; C57BL/6 *prf*$^{-/-}$ $n = 6$ and BALB/c $n = 5$ 10 weeks old male mice in all). Boxplot diagrams show min, max, median 1st and 3rd quartil. *$p < 0.05$, **$p < 0.01$, ***$p < 0.001$, two-tailed unpaired heteroscedastic *t*-test.

pathology we included in our analysis *Pb*NK65-infected C57BL/6 wt mice that are known to not develop cerebral malaria[27]. Here, as studied on day 7 p.i., we also did not find the black-brown deposits present in the iRBCs of *Pb*A-infected C57BL/6 wt mice (Fig. 2S2A–E). In all three models we found black-brown deposits in bone marrow cells arguing for skull bone marrow infections (Fig. 4J, T, S2C). Bone marrow infections in murine models have been described earlier[31]. The infection of the murine skull bone marrow, however, is to the best of our knowledge a novel finding.

At day 5 p.i., which was the earliest time point examined, we found in *Pb*A-infected C57BL/6 wt mice iRBCs in the large draining veins, either single iRBCs among non-infected RBCs (Fig. 5M, N, O) or densely packed groups of iRBCs (Fig. 5J, K, L) and also iRBCs in direct contact to the vessel walls (Fig. 5E, F, G). The same types of deposits that we detected in these iRBCs, *i.e.*, dark-brown granules that polarized light, were also found in parts of the vessel walls (Fig. 5F, G) suggesting vessel reactions and sequestration of iRBCs in the large draining veins, or certain compartments of these veins (Fig. 5C).

In conclusion, our histological results confirm the findings of our iRBC-SPECT-imaging study. IRBCs are present in large draining veins, and vessel wall reactions are indicative of iRBC sequestration.

**MR-angiography and histology uncover two parallel venous streams from the superior sagittal sinus to the rostral rhinal veins**

In accordance with our TOF-MRA findings we also noted in our histological study that the sss, at around Bregma +0.6 mm splits into two vessels positioned on top of each other. Histologically and in X-ray CT images we find that the upper vessel runs within the diploic space and rostrally divides into left and right rrv (Fig. 6A–H) which have direct connections to the bone marrow (Fig. 6J). The lower vessel can be seen as the superior sagittal sinus proper as it runs below the bone into the rostral confluence of the sinus. Here, both venous streams are connected again (Fig. 6G). The sagittal TOF-MRA images (Fig. 2, BALB/c d0) in conjunction with a low-density area in the CT (Fig. 6E) indicate

that a third connection exists between both streams in the caudal one-third of both vessels. In particular in the upper stream (Fig. 5C; 6P, Q, S, T) but also in the lower (Fig. S2F) we find vessel wall reactions in infected C57BL/6 wt mice.

Our combined histological, X-ray CT, and TOF-MRA study thus uncovered a vascular anatomy of the rostral superior sagittal sinus that is more complex than the traditional scheme. Two interconnected parallel venous streams instead of one single sinus run rostrally towards the rostral confluence of the sinuses and the rostral rhinal veins.

**Impaired perfusion in territories of large draining cerebral veins in *Pb*A- infected mice**

In order to study the impact of early iRBC sequestration in the brain and the large draining veins on cerebral perfusion, we used $^{99m}$Tc-hexamethylpropylene amine oxime ($^{99m}$Tc-HMPAO) SPECT, a technique widely used in clinical routine for imaging spatial patterns of CBF. We imaged uninfected and infected C57BL/6 wt mice at day 5 p.i. Notably, we did not find in any individual perfusion deficits indicating arterial infarctions, *i.e.*, severely reduced flow in territories of the major arteries or branches thereof. Instead, we found a moderate decrease in cerebral perfusion, which was largely bilaterally symmetric and most prominent in the dorsocaudal olfactory bulb, the rostral prefrontal cortex, the caudal midline cortex, the adjacent visual cortices as well as in upper midline superior colliculus (Fig. 7). These regions are directly underlying or adjacent to the junctions of the sagittal sinus with the rostral rhinal veins and the transverse sinus, respectively, where the iRBC accumulation was most prominent (Fig. 1). This finding indicates early alterations in perfusion in territories of these large draining veins. We noted additional decreases in CBF in the hippocampal formation and entorhinal cortex (Fig. 7, Bregma −3.3), areas that finally drain into the straight sinus[32]. The tendency towards bilateral alterations would be in line with venous efflux reduction in an unpaired midline vessel playing a causal role. However, due to the small isolated foci of reduced CBF, the congruences of affected areas and venous drainage

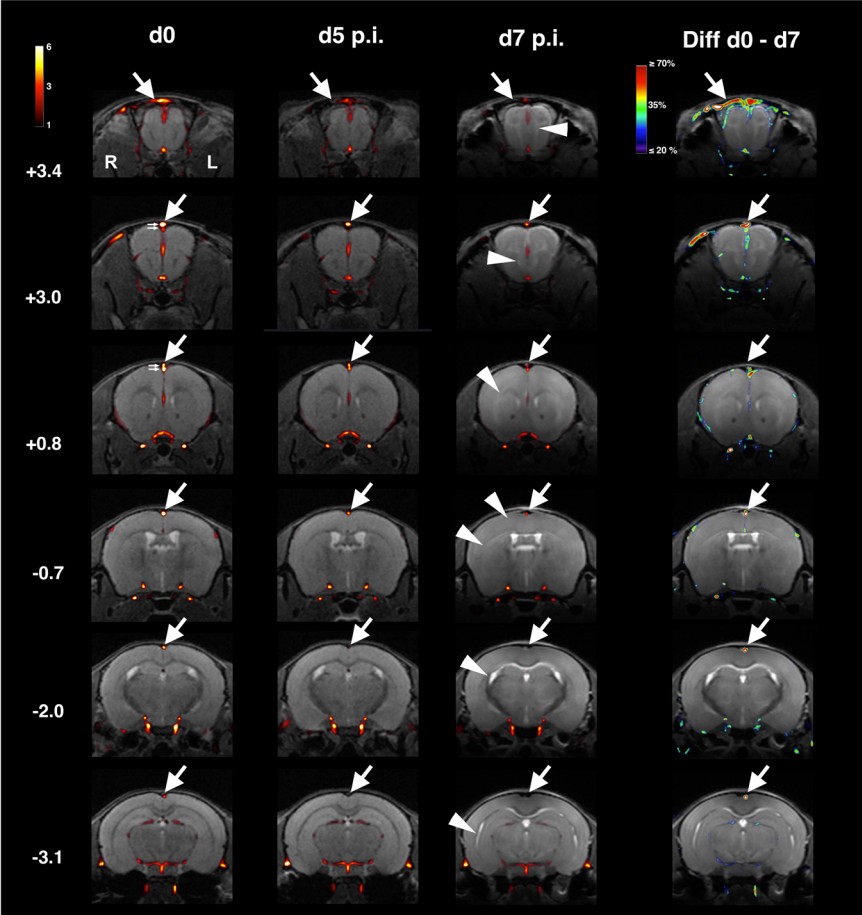

**Fig. 3 | In *Pb*A-infected C57BL/6 wt mice flow is reduced most severely in the superior sagittal sinus and the rostral rhinal vein.** Rostral to caudal series of sections of group-mean MR-angiographies from mice at d0, d5 p.i. and d7 p.i. overlaid on T2-MR-images from single individuals from the according groups in C57BL/6 wt mice and reduction in mean signal intensities between d0 and d7 p.i. Color scales are in signal intensities of the TOFs and in percentage difference in Diff d0-d7. Bregma coordinates are given on the left side. Signal intensities decrease most severely in the rrv (arrows in the uppermost row, Bregma 3.4) and the sss (arrows in Bregma +3.0 to −3.1). Peak reductions in signal intensity in the sss are larger than 70%. Edema is particularly prominent in white matter (arrowheads). Note also the anatomical relationships with respect to the sss (small arrows in d0 at Bregma levels +3.4 and +0.8). There are two vessels, the inferior one is the sss, the superior one originates from the sss and opens into the rrv.

territories are less apparent than in the olfactory bulb/rostral rhinal vein region.

### ECM-associated brain pathology and pro-inflammatory responses are late events in ECM pathogenesis

In order to relate the in vivo imaging findings to brain region-specific neuroinflammation and pathology, we studied the brains of C57BL/6 wt mice at corresponding time points using an array of histological and molecular techniques. Very few signs of brain pathology were visible at day 5 p.i. (Fig. 8A–F), which progressed until day 7 p.i., where the mice developed widespread neuroinflammation. Interestingly, the intracerebral infiltration of CD8+ T cells was negligible at day 5 p.i. and similar to uninfected mice (Fig. 8D). However, numbers of intracerebral CD8+ T cells were significantly increased at day 7 p.i. in all brain regions, predominantly in the sub-ventricular zone (36-fold) and midbrain region (33-fold) (Fig. 8D, Fig. S7). Thus, CD8+ T cells, which contribute unambiguously to the lethal outcome of ECM, increase only after the onset of iRBC accumulation and reduced venous efflux in the brain.

This finding prompted us to analyze the spatial and temporal expression of intracerebral pro-inflammatory cytokines and chemokines in the olfactory bulb, cortex, and brainstem (Fig. 9). At day 5 p.i., when there was a significant accumulation of iRBCs in the brain

(Fig. 1K), mRNA levels of the pro-inflammatory cytokines IFN-γ, and TNF and the chemokines CXCL9, CXCL10, CXCL11, which are essential for the recruitment of pathogenic CD8+ T cells to the brain, were only slightly induced. In marked contrast, at day 7 p.i. there was a significant increase in levels of the chemokines and cytokines throughout all three brain regions of *Pb*A-infected mice (Fig. 9).

In good agreement with the increase of CD8+ T cells and upregulation of cytokines and chemokines from day 5 to day 7 p.i., the numbers of GFAP+ astrocytes (Fig. 8A, S4) and Iba1+ microglia (Fig. 8B, S5) and CD31/PECAM-1+ staining intensity on endothelial cells (Fig. 8C, S6) increased in all brain regions gradually from day 0 to day 7 p.i. Of note, the strongest increase was again from day 5 to day 7 p.i. further illustrating that neuroinflammation and activation of brain resident cell populations developed later than significant accumulation of iRBCs in cerebral blood vessels. Interestingly, as observed in human CM, in our murine ECM model large hemorrhagic lesions in the brainstem were observed (Fig. 8F, S8), which also stained positive for cleaved caspase-3, an apoptosis-inducing effector caspase (Fig. 8E, S9), reproducing CM pathology in ECM. Hemorrhagic lesions and apoptotic cells emerged only later at day 7 but not at day 5 p.i.

Taken together, our data show that neuroinflammation is a late event and develops after significant accumulation of iRBCs in cerebral blood vessels in ECM.

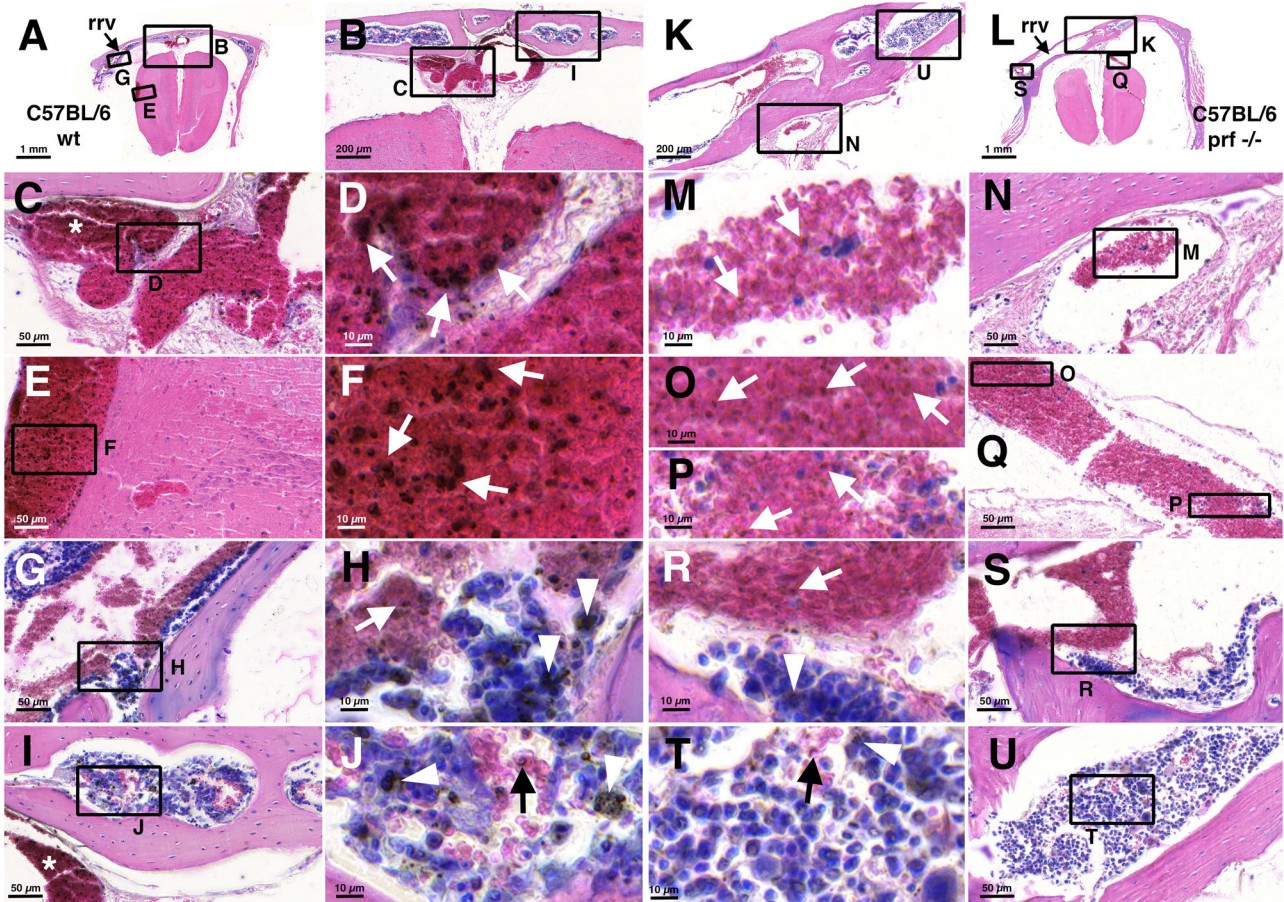

**Fig. 4 | Large deposits accumulate in iRBCs in draining veins at day 7 after *Pb*A-infection in C57BL/6 wt but not in C57BL/6 *prf*⁻/⁻ mice.** Giemsa-stained frontal sections at the level of the rrv, from *Pb*A-infected mice at d7 p.i., C57BL/6 wt in the two columns on the left, C57BL/6 *prf*⁻/⁻ in the two columns on the right. Overviews are shown in (**A**) and (**L**) resp., details from (**A**) and (**L**) in (**B**) and (**K**). Arrows in (**A**) and (**L**) point to the rrv. Details at medium magnification are shown in (**C, E, G, I** and **N, Q, S, U**) resp., high-resolution images from these details in corresponding adjacent columns (**D, F, H, J** and **M, O, P, R, T**). Frames in lower-resolution images indicate the positions of details in higher-resolution images. Large midline veins at or close to the rostral confluence of the sinus (rcs) are shown in (**C**) and (**N**). Note in the wt mouse (**C**) the accumulations of large-diameter dark-brown/black deposits in the RBCs in the large veins in the area of the rcs (**C, D**), in a vein draining the olfactory bulb (**E, F**), in the rrv (**G, H**) and in the bone marrow (**I, J**). IRBCs can be found in all large draining veins in the C57BL/6 *prf*⁻/⁻ mouse (arrows in **M, O, P, R**) but the large-diameter deposits visible in the wt mouse are absent. Deposits are present also in white blood cells in the rrv (arrowheads in **H** and **R**) and in bone marrow cells (arrowheads in **J, T**), in both cases less pronounced in the C57BL/6 *prf*⁻/⁻ mouse.

### Early anti-malaria treatment reverses brain pathology

Since our data support the notion that intracerebral iRBC accumulation is essential for ECM development, we next wanted to validate whether the pro-inflammatory response was a consequence of iRBC accumulation. To this end, we treated infected C57BL/6 wt mice with the anti-malarial drug pyrimethamine starting from day 5 p.i. Anti-malarial treatment rapidly reduced the amount of iRBC within 24 h (Fig. S10A) and resulted in survival of all mice (Fig. S10B). Furthermore, BBB damage in brains of pyrimethamine-treated mice was absent in contrast to untreated animals (Fig. S8C).

Importantly, treatment with the antimalarial drug reduced brain pathology (Fig. 8) and resulted in low pro-inflammatory cytokine and chemokine responses at day 7 p.i. (Fig. 9). These findings indicate that anti-malarial treatment at the time of intravascular accumulation of iRBC can prevent the complications leading to the pathogenesis of ECM suggesting that sequestration of infected red blood cells and reduced venous efflux precede inflammation in experimental cerebral malaria.

### Disruption of the rostral migratory stream is a consequence of iRBC accumulation

We included in our study the analysis of the rostral migratory stream (RMS) and replicated the findings of Hoffmann et al. [29]. in C57BL/6 wt mice. The migration and proliferation of neuroblasts in the rostral migratory stream (RMS) were disturbed already at day 5 p.i. as indicated by reduction in the number of doublecortin (DCX) and bromodeoxyuridine (BrdU) stained neuroblasts (Fig. S3A, B). The disruption progresses further until day 7 p.i., where the RMS was completely disturbed.

Interestingly, early treatment not only prevented the damage of the RMS, but also fostered its repair, indicating that the disruption of the RMS is consequence of RBC accumulation (Fig. S3A–E).

### Discussion

In the present study we analyzed, for the first time, the 3D in vivo distribution of iRBCs in experimental cerebral malaria at an early stage of the disease, where neurological symptoms, inflammation, and edema are virtually absent. Using SPECT-imaging in a murine ECM model we found significant increases of ⁹⁹ᵐTc-labeled iRBCs in the brains and in large draining veins or sinuses. Simultaneously, venous efflux in large cerebral draining veins and sinuses, as determined by TOF-MRA, decreased at day 5 p.i. and was further reduced to nearly undetectable levels in symptomatic affected mice at day 7 p.i. SPECT-imaging of CBF using ⁹⁹ᵐTc-HMPAO as a tracer revealed moderate decreases in perfusion at day 5 p.i. Affected regions were found in the

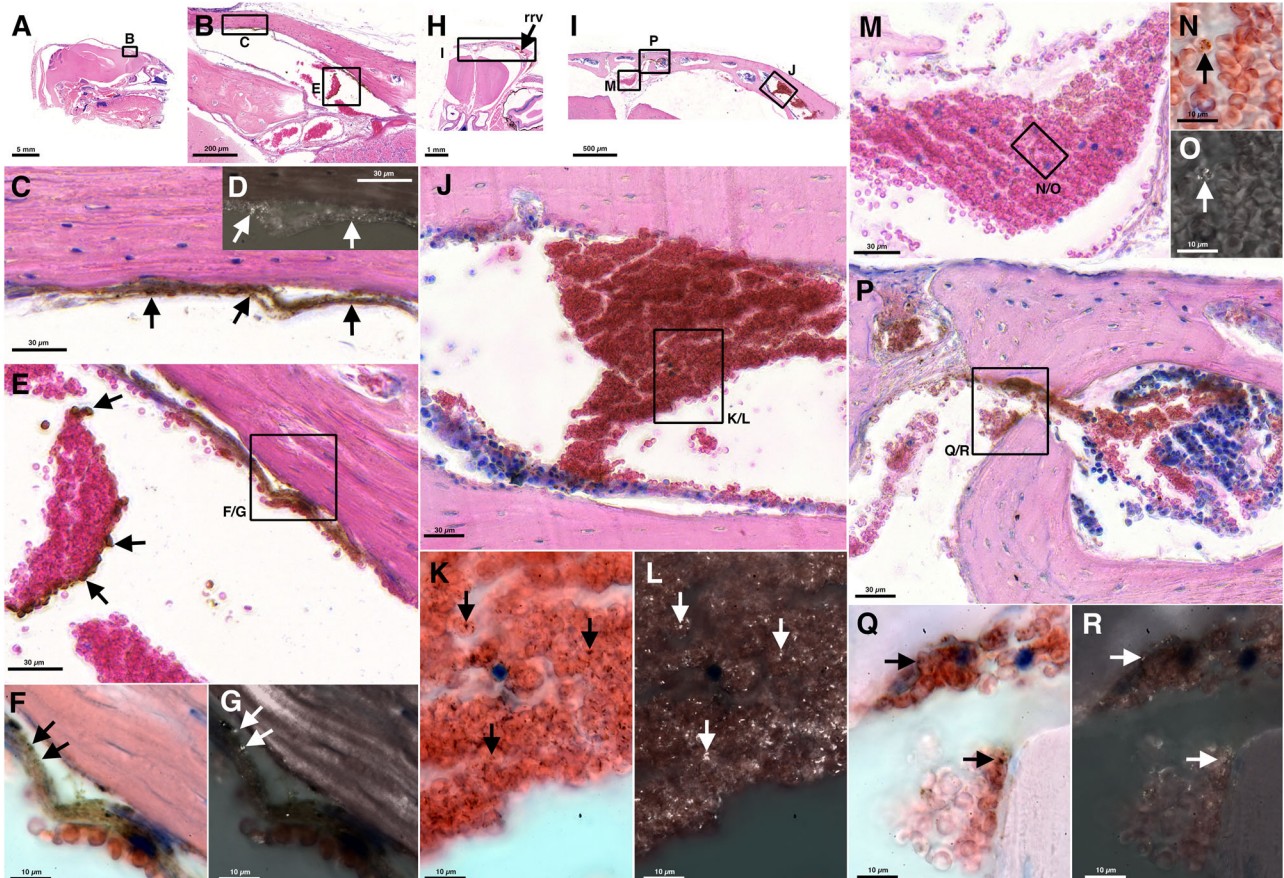

**Fig. 5 | At day 5 p.i. large draining veins in C57BL/6 wt mice show vessel wall reactions and sequestration of iRBCs.** Giemsa-stained midsagittal (**A**–**G**) and frontal section at the level of the rrv (**H**–**R**). Frames in lower-resolution images indicate positions of higher-resolution details. Images from oil immersion microscopy under non-polarized vs. polarized light are shown in (**F**, **K**, **N**, **Q**, and **G**, **L**, **O**, **R**) resp., a lower resolution comparison in (**C**, **D**). Images under polarized light are black and white images overlaid with opacities from 80–90% on images under non-polarized light. Note that focal planes of maximum emission of polarized light and maximum absorption under non-polarized light can differ slightly. Endothelial vessel linings close to the entrance of the sss into the rrv (**C**, **D**) and in the region of the rcs (**F**, **G**) show a dark-brown staining (black arrows) and polarizing material (white arrows). IRBCs are found within the vessels (arrows in **E**, **J**, **M**), at relatively high density within the rrv (arrows in **I**, **J**, **K**). IRBCs are present in the bone marrow and in or close to channels connecting bone marrow and draining veins (arrows in **Q**, **R**).

caudal olfactory bulb, rostral prefrontal cortex as well as in the posterior cingulate and visual cortex, areas from which venous efflux drains into the rostral rhinal veins or the superior sagittal sinus, respectively[25,32,33].

The 3D distribution of iRBCs as determined with SPECT is in excellent agreement with previous 2D optical ex vivo whole-brain images of the parasite distribution, obtained with luciferase-expressing *Pb*A[19,20,34]. Although the relationship of the parasite distribution to the venous sinuses has remained unnoticed, the published data can now be readily interpreted as 2D projection images of the 3D distribution described in our study.

We independently confirmed the presence of iRBCs in the large draining veins histologically. We note that in a previous comprehensive study of the distribution of GFP-labeled parasites in the murine brain the sinuses were not included[14]. We propose that attention to parasite distribution to the venous sinuses and omission of transcardial perfusion will be important for future studies on the dynamics of ECM onset.

As early as day 5 p.i., we find in *Pb*A-infected C57BL/6 wt mice not only inside the iRBCs but also in vessel walls black-brown deposits polarizing light and thus most likely representing hemozoin. This suggests a vascular or endothelial reaction to iRBCs as described in the literature[35] and iRBC sequestration. At day 7 p.i. we find large-diameter deposits similar in size and shape to those of rosetting iRBCs, as described before[36].

To substantiate our findings, we also included two murine systems, *Pb*A infections in C57BL/6 *prf*⁻/⁻ and BALB/c mice, which develop severe anemia after protracted parasitemia but do not develop ECM. Venous efflux was functional and large-diameter deposits were not detectable in these infection models. In agreement with earlier results from MR-imaging[30] infected C57BL/6 *prf*⁻/⁻ mice did not develop edema. Our findings suggests that vessel pathology, as shown by deposition of parasitic material in the vessel wall, requires contribution of the immune system, likely by activating endothelial cells, a process in which perforin might be involved. We also note that edema was only detectable after flow in the large draining veins was substantially compromised.

Our interpretation is supported by a recent study[37] on protective effects of hypothyroidism in ECM. In hypothyroid C57BL/6 wt mice, *Pb*A infections did not result in increased intracranial pressure and edema despite inflammation and BBB-breakdown. In these mice the flow in the large vessels, as determined with TOF-MRA, was maintained. Although angiographies were not analyzed at the single-vessel level, the flow in the sinuses is clearly visible in these TOF-MRA images[37].

Together, these findings challenge the traditional view of edema generation by pathological alterations in the microcirculation and are

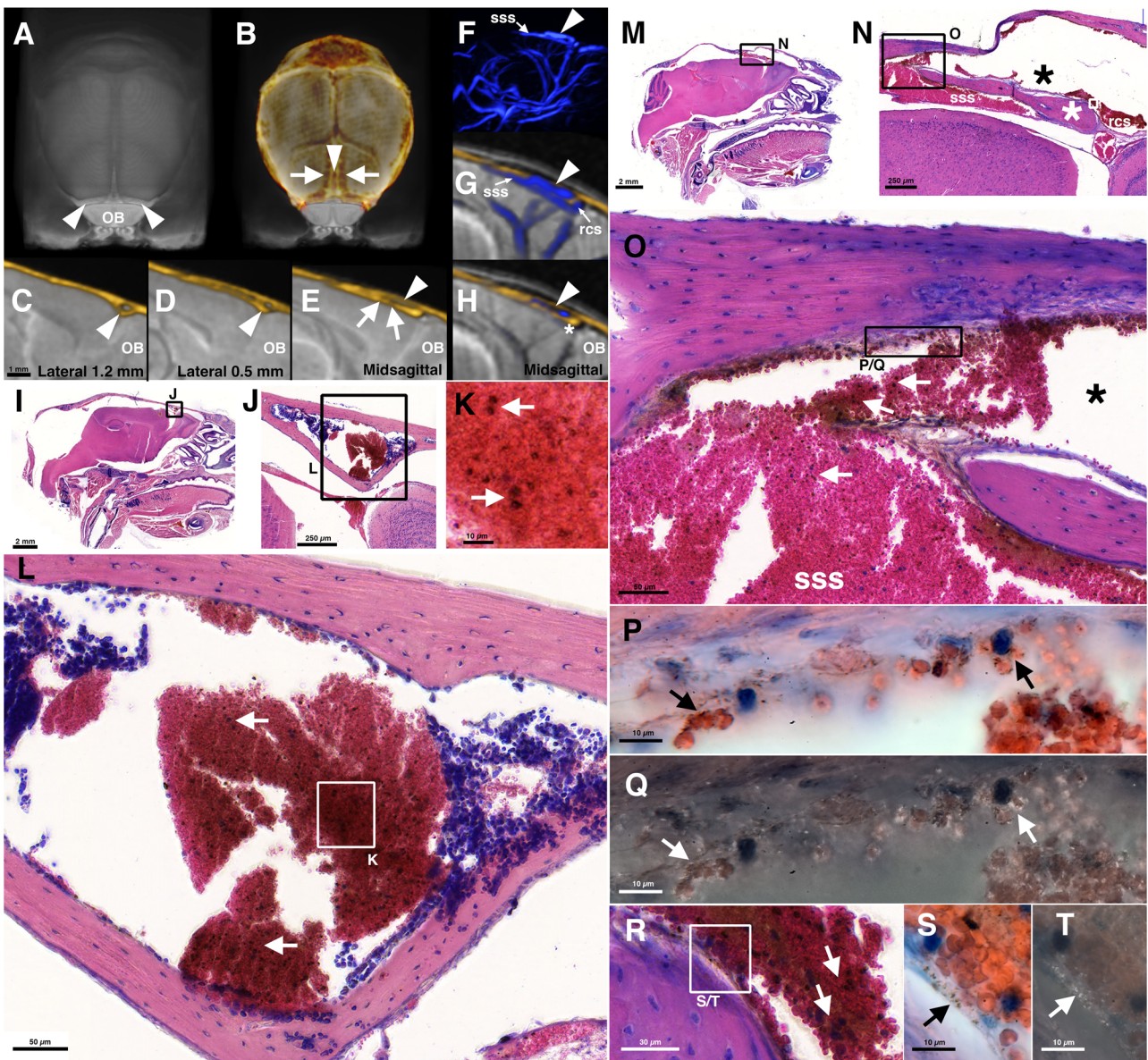

**Fig. 6 | Topographical relationships of major sites of pathology, the superior sagittal sinus and the rostral rhinal vein.** Shown are MR-images from a contrast-agent perfused ex vivo brain[76] (**A**–**E**) with an overlay of a CT (yellow in **B**–**E**), T2w MR, and TOF-MR-angiographies from an uninfected individual (blue in **F**, **G**, **H**) with an overlay of the same CT (**G**, **H**) as well as images from a parasagittal (**I**–**L**) and a largely midsagittal (**M**–**T**) paraffin section of a C57BL/6 wt *Pb*A-infected mouse at day 6 p.i. at the level of the rrv and the sss respectively. A top view on the rrv is shown in (**A**) (arrowheads above the olfactory bulb OB). In the overlay of the cranium (**B**) three compartments of lower bone density can be distinguished behind the rrv, a midline compartment (arrowhead), and two compartments on each side (arrows). The midline compartment as demonstrated in a midsagittal section (arrowhead in **E**) harbors a venous space (arrowhead in **H**) that is connected to the sss by openings in an osseous lamella (arrows in **E**). In 3D-renderings of TOF-MRs this venous space can be identified above the sss (arrowhead in **F**). The borders between the sss and the upper compartment may be difficult to discern (**G**). Images of peak TOF signal intensities can show the isolated compartment (**H**). In the midsagittal paraffin section, the compartment is marked by a black asterisk in (**N** and **O**). The compartment is artificially enlarged due to a distortion of the overlaying decalcified bone. Rostrally it is connected to the rcs (**G** and **N**). The club-shaped bone caudal to the rcs (white asterisk in **N**) can be identified in midsagittal CT images (above the white asterisk in **H**). The rrv is a continuation of the midline venous space (**J**, **L**, and arrowheads in **C**, **D**). The entire space is densely filled with iRBCs (arrows in **K**, **L**, **O**, **R**). Details under polarized (**Q**, **T**) or non-polarized (**P**, **S**) light show deposits not only within iRBCs (**P**, **Q**) but also in the vessel linings (**S**, **T**).

fully compatible with an alternative concept of edema generation as a consequence of downstream reductions of venous efflux. We propose that the spatiotemporal patterns of edema development, and of ECM pathology in general, are triggered by reduced efflux in the large draining veins and later exacerbated by local pathological alterations within the microcirculation.

In rodents, the superior sagittal sinus drains into the rostral rhinal veins[25,33] forming a major venous efflux route that differs from humans. In *Pb*A-infected C57BL/6 mice, we find the highest intensity of labeled iRBCs at the T-shaped junction of the sagittal sinus and the left and right rostral rhinal veins. The earliest signs of pathological alterations occur in regions draining into these veins and earliest signs of blood-brain barrier disruptions are detectable in brain tissue directly below the junction. A previous MR-study showed the first signs of edema and blood-brain barrier disruption in the caudal olfactory bulb, a region also severely affected by hemorrhages at later stages[29]. From there the edema spreads caudally along white matter fiber tracts affecting, among other structures, the corpus callosum. The spatiotemporal dynamics of edema onset and progression are unlikely to be influenced by host factors governing iRBC densities in the microcirculation.

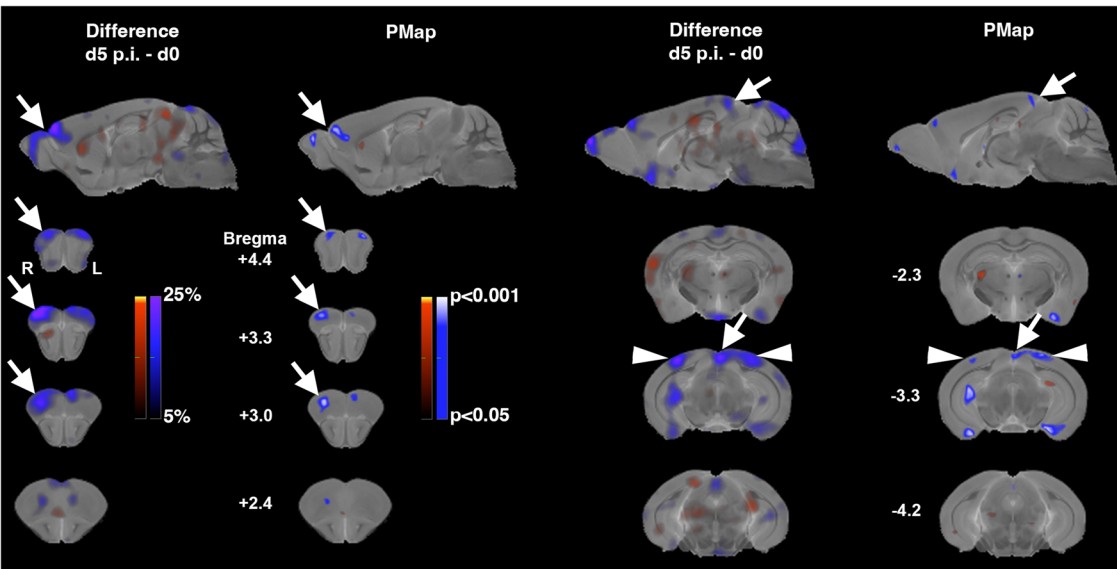

**Fig. 7 | Cerebral blood flow is impaired early in territories of large draining veins.** Spatial patterns of the differences in cerebral blood flow (CBF) between infected mice at day 5 p.i. ($n = 9$, male) compared to controls ($n = 10$, male). Percentage differences in CBF as determined in global-mean normalized [99m]Tc-HMPAO SPECT data are shown in the first and third column (d5 p.i. minus d0), results from a voxel-wise unpaired heteroscedastic t-test (p-values, uncorrected) in the same sections in the second and fourth column (PMap). Hot and cold color scales denote increases and decreases, respectively. Images are overlaid on an anatomical reference MR[76]. Parasagittal sections slightly right (left half of the figure) and left (right half of the figure) of the midline are shown in the uppermost row, frontal sections at indicated Bregma levels in the lower rows. Differences are most pronounced in caudal olfactory bulb and rostral prefrontal cortex (arrows pointing left downward), with decreases in CBF up to 25% in infected mice. CBF is also decreased in posterior cingulate cortex and upper midline superior colliculus (arrows pointing right downward) as well as bilaterally in the visual cortex.

Instead, impaired flow in the superior sagittal sinus and the rostral rhinal veins provide a plausible explanation for edema dynamics and parasite distribution in the brain. According to this model, differences between mice and humans in the spatial patterns of edema onset and progression can be readily explained by differences in venous efflux anatomy. Together, our data are consistent with a model that edema formation is causally related to reductions in flow in the large draining veins, which in turn are caused by the accumulation of iRBCs.

Given the high sensitivity and prompt reactivity of the intracranial pressure to venous efflux obstructions[38] it seems possible that iRBC accumulation is a major factor leading to edema and brain death, particularly in cases with high parasite load. In infected patients, correlations were observed between brain swelling and parasite molecular markers in blood plasma related to iRBC biomass[39], while peripheral plasma cytokine levels did not correlate with brain swelling[40]. Intriguingly, *post-mortem* studies have identified a subgroup of patients in which iRBC sequestration was the only pathological finding[9]. We hypothesize that iRBC accumulation in large draining veins and venous efflux obstructions are unifying hallmarks of onset of pediatric CM and ECM in murine models. Recent findings of increased hemoglobin concentrations, most likely representing vascular congestion in the microvasculature, as measured with near-infrared spectroscopy in infected children in vivo[41], support this assumption and could represent a direct consequence of downstream obstructions.

Thus, inflammation accompanied by cytokine and chemokine production and CD8[+] T cell infiltration are downstream events serving as robust pathological read-outs in ECM studies, but not necessarily being critical disease signatures in pediatric CM. Most importantly, recognition of reduced venous efflux in conjunction with iRBC sequestration lend strong support to the notion that murine malaria models can mimic CM and further emphasize the need to study disease onset prior to apparent clinical symptoms.

The effects of impairments in venous efflux in large draining veins are well known from cerebral venous thrombosis and mechanical obstructions of sinuses in both humans[38,42] and animal models[43–45].

Hydrostatic pressure rises in upstream venous and capillary vessels and - depending on the presence or absence of collaterals - disruption of the blood-brain barrier, decreased perfusion, edema, and hemorrhages may follow.

Our data do not reveal whether the reduced efflux in the large draining veins was accompanied by thrombosis. In adult humans, sinus thrombosis in CM has been reported[46–48], but this is generally regarded a rare event[46]. In African children, sinus thrombosis has neither been reported in *post-mortem*[11] nor in MRI studies[4]. The apparent lack of sinus thrombosis, however, does not preclude impairments of blood flow. This could be caused by combinations of iRBC sequestration, rosetting, and reduced deformability, which together likely impair fluid dynamics and lead to rheopathological effects that have been discussed earlier as major factors in CM pathology in both mice and humans[49,50]. An MRI-study in Malawi children explicitly mentioned that transient occlusion of the internal cerebral vein and the vein of Galen could explain the pathology of a subgroup of patients[4]. We note that sequestration alone might not be sufficient to reduce the flow in large vessels to critical levels for inducing an edema. Additional alterations in viscosity and fluid dynamics are likely required, and the differences in flow observed between *Pb*A-infected C57BL/6 wt and *prf*[−/−] mice might be attributed to cofactors, such as rosetting and hemozoin load.

There is broad consensus that the pathophysiology of CM is complex, and there is no one single disease signature in human *post-mortem* material or in neuroimaging in humans[15]. Thus far, studies and theories on CM pathology have focused on the microcirculation. Intravascular sequestration is the prime mechanism for iRBCs to avoid clearance by the spleen[51], and of all vascular segments the capillary networks provide the largest surface area for sequestration. iRBC sequestration and/or rosetting in both the capillary and the venous space might be mediated by distinct or shared parasite-encoded proteins on the surface of infected erythrocytes. For instance, a member of the *Plasmodium falciparum* erythrocyte membrane protein 1 (*Pf*EMP1) family that binds to endothelial protein C receptor (EPCR) has been linked to brain swelling in pediatric cerebral malaria[39], but

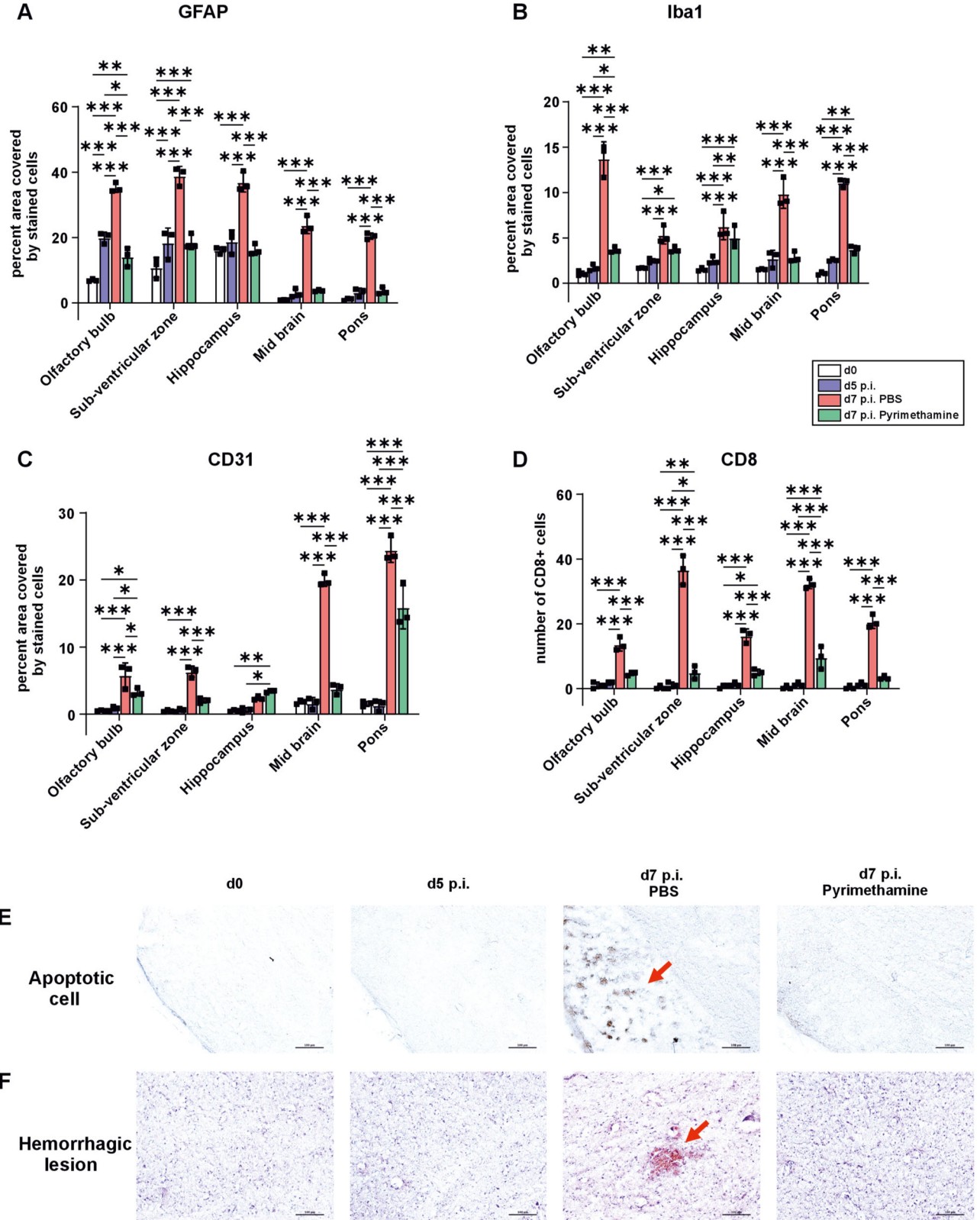

**Fig. 8 | ECM-associated brain pathology progressively worsens and is not region-specific.** Histopathology was performed on brains isolated from 10-week-old female uninfected mice (day 0), day 5, and day 7 p.i. **A** group of infected mice was also treated with pyrimethamine and examined on day 7 after parasite inoculation. **A**, **B** Increased activation of astrocytes (**A**) and microglia (**B**) in the brains of day 7-infected mice compared to controls and treated mice, as shown by glial fibrillary acidic protein (GFAP) and ionized calcium-binding adapter molecule 1 (Iba-1) staining, respectively. **C** CD31 staining shows increased activation of endothelial cells in the brain of day 7-infected mice. **D** Accumulation of CD8⁺ T cells in the brain of day 7-infected mice. **E**, **F** Hemorrhagic lesions (red arrows), viewed by H&E staining (**F**), and apoptotic cells, viewed by cleaved caspase-3 staining (**E**), were observed in the brainstem regions of day 7-infected mice. Early anti-malarial treatment protects the brain and recovers the brain pathology in all the regions. $n = 3$ independent animals per group for **A**–**D**. $*p < 0.05$; $**p < 0.01$; $***p < 0.001$ (two-way ANOVA). Data are presented as mean values ± SD.

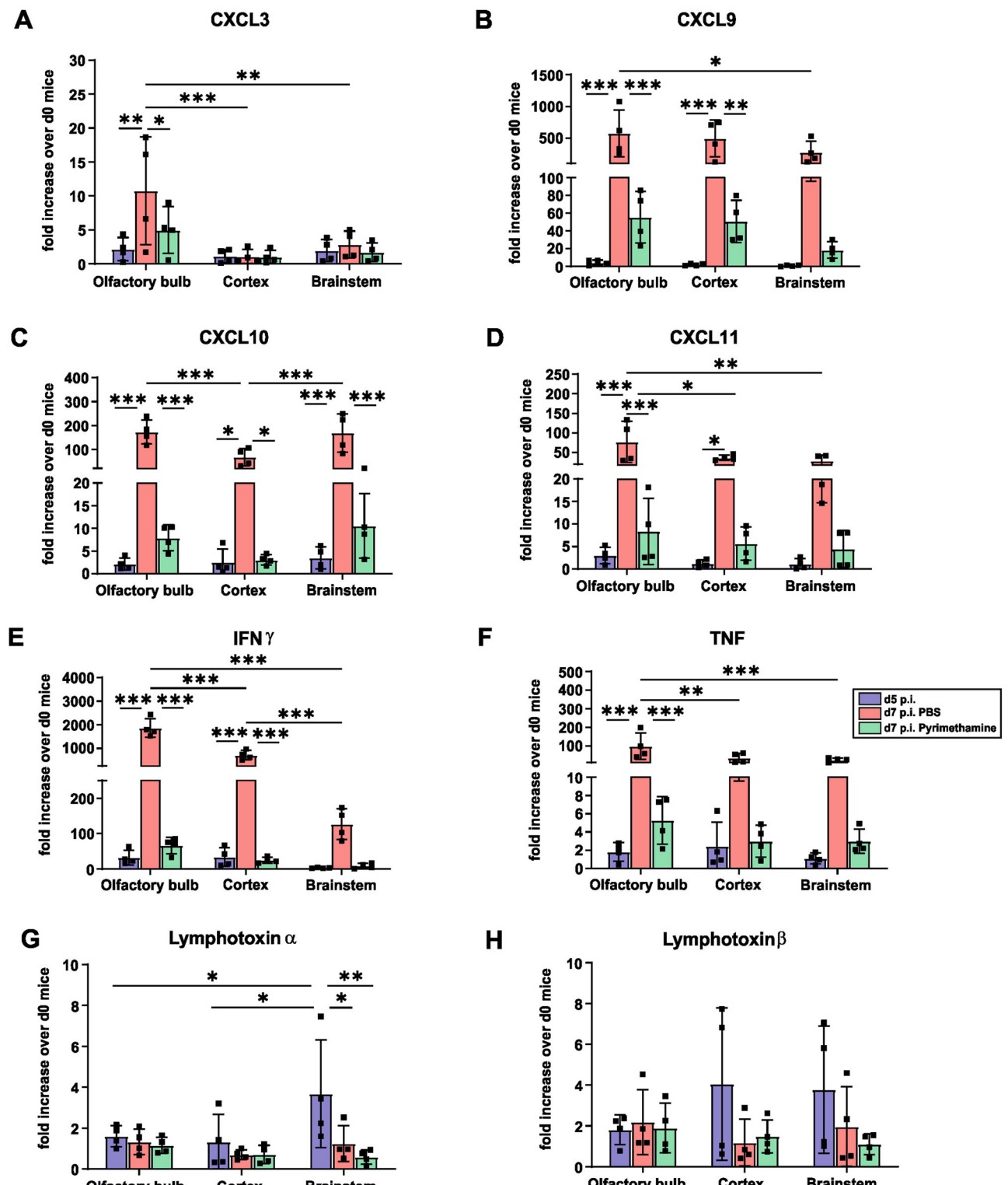

**Fig. 9 | ECM augments cytokine and chemokine production in the later stages of the disease.** Quantitative real-time polymerase chain reaction analysis of intracerebral chemokines CXCL3 (**A**), CXCL9 (**B**), CXCL10 (**C**), and CXCL11 (**D**) and pro-inflammatory cytokines interferon-γ (IFN-γ) (**E**), tumor necrosis factor (TNF) (**F**), lymphotoxin-α (Ltα) (**G**) and Ltβ (**H**) by quantifying the mRNA expression in the olfactory bulb, cortex, and brainstem of day 5 p.i., day 7 p.i. pyrimethamine-treated and untreated mice (9-week-old female mice). Data show significant increase in mRNA expression of IFN-γ, TNF, CXCL9, CXCL10, and CXCL11 in three different brain regions of day 7-infected mice and a substantial decrease upon anti-malarial treatment. $n = 4$ independent animals per group for all. *$p < 0.05$; **$p < 0.01$; ***$p < 0.001$ (two-way ANOVA). Data are presented as mean values ± SD.

potential candidates in murine malaria remain enigmatic. Clearly, future studies on the molecular mechanisms and parasite-host interactions during iRBC sequestration in large draining cerebral veins are needed. There is precedence in the human retina, where iRBC sequestration in large veins has been clearly demonstrated[52]. Several studies have shown that malaria retinopathies can indicate cerebral malaria in infected neurologically symptomatic patients[53,54], and it will be interesting to explore whether iRBC sequestration in large veins is a contributing factor in addition to occlusion of the retinal microvasculature and hypoxic injury. Of note, a similar sequestration phenotype has also been described in larger pial and subarachnoid vessels in human *post-mortem* material[10]. These observations together with our findings in the *Pb*A-infected C57BL/6 wt mice suggest that a systematic analysis of the iRBC content in large veins in human *post-mortem* brains is warranted.

Edema in pediatric CM, as detected with MRI, is typically bilaterally symmetrical, affecting larger contiguous areas that do not correspond to arterial territories. It can be preferentially cortical with anterior or posterior phenotypes, subcortical or global[4,5,55,56]. These diverse phenotypes can now be readily explained by venous efflux reductions in large draining veins with some individual variations in onset sites of critically reduced flow, i.e., superficial veins including bridging veins, sinuses, deep veins or globally affected large veins, rather than by varying predilection sites within the host microcirculation.

Especially in case of limited collateral flow, venous watershed areas could be particularly prone to damage, a phenomenon highlighted in human retinal pathology[57]. It is of particular interest that the hemorrhages in children who died of CM are found in a confined subcortical white matter zone in *post-mortem* brains[9]. This subcortical white matter zone, which is also a well-delineated zone of corresponding MR-pathology in vivo[55], is a prominent venous watershed area between deep and superficial veins, an area also known for the so-called juxtacortical hemorrhages in cerebral venous thrombosis (see Fig. 4 in Coutinho et al. [58]).

In the murine malaria model, important events associated with ECM pathogenesis are intracerebral pro-inflammatory cytokine responses and influx of pathogenic CD8+ T cells[59–61]. In good agreement, our study shows a progressive increase in cytokine levels starting at day 5 p.i. and peaking at day 7 p.i. when the mice succumb to the disease. The increase in cytokine levels was accompanied by an increase in a pro-inflammatory response, characterized by activation of brain parenchymal cells and infiltration of disease-causing CD8+ T cells, ultimately resulting in the disruption of the BBB. Anti-malarial treatment at day 5 p.i., when the peripheral parasitemia was already established, was sufficient to dampen the pro-inflammatory responses, indicating that accumulation of iRBC was essential for intracerebral pro-inflammatory response. To what extend cytokines occur in pediatric CM is not firmly established, largely due to the limitation that the measurements can only be performed in *post-mortem* tissue.

Notably, we detected iRBCs in the skull bone marrow. There is now consensus that bone marrow sequestration by iRBCs is a major site for parasite accumulation, gametocyte maturation, and, perhaps, parasite persistence in *Plasmodium falciparum* and *Plasmodium vivax* infections[62–64]. Accordingly, iRBC accumulation in the skull bone marrow is expected, and we provide the first experimental evidence in a murine ECM model.

Our combined SPECT/CT and MR-angiography approach led to the identification of a previously unrecognized venous anatomy of the rostral superior sagittal sinus region, i.e., two interconnected parallel venous streams. Images in a number of studies from recent years indicate that this region is one of the nodal points of the lymphatic[65] and glymphatic[66] drainage systems and involved in bone marrow–brain interactions[67,68]. The venous system in this area might,

thus, share functional similarities not only to the *superior sagittal sinus proper* in humans but also to human perisinus spaces, *e.g., venous lacunae*, which are connected to diploic veins, as well as arachnoid granulations, in which recently lymphatic conduits and communication routes to the bone marrow have been demonstrated[69]. These lacunae in humans are relatively large venous spaces that might be well-suited for iRBC sequestration or passive "sedimentation". This could impair venous efflux through the bridging veins and might result in severe edema even with partially intact flow within the core of the superior sagittal sinus.

In summary, we show early stage iRBC sequestration and reduced flow in large draining veins only in *P. berghei*-infected brains that are on a trajectory to ECM. We propose that in CM and ECM pathology edema and hemorrhage formation are causally linked to iRBC-induced reductions in venous efflux in large draining veins downstream of the microcirculation and postcapillary venules.

## Methods

### Ethics statement
All animal experiments were in compliance with the German Animal Welfare Act (TierSchG) in a protocol approved by the state authorities (Landesverwaltungsamt Sachsen-Anhalt) file no: 42502-2-1440 UniMD.

### Animals
Three different mouse strains were used in this study, C57BL/6 wt mice obtained from Janvier (Le Genest Saint Isle, France) and Jax, perforin-deficient C57BL/6 mice (C57BL/6-Prf1tm1Sdz/J, https://www.jax.org/strain/002407) and BALB/c mice (https://www.jax.org/strain/000651) obtained from Jax. All animals were kept in isolation under pathogen-free conditions in groups of 4 in standardized cages with 12 h of light-dark cycle and food and water was provided *ad libitum* in the animal facility of the Otto-von-Guericke University, Magdeburg, and the Leibniz Institute for Neurobiology, Magdeburg. For SPECT-imaging, mice were single-housed after implantation of the jugular vein catheter.

### Parasite infection
The reference line 'cl15cy1' of the ANKA strain of *P. berghei* was used for the infection[70]. For blood-stage infection, parasites were passaged in C57BL/6 mice ($n = 20$). At day 7 p.i., blood was collected through cardiac puncture, and blood stabilates were prepared by mixing 100 µl blood containing of $3 \times 10^6$ iRBCs with 200 µl Alsever's solution (Sigma–Aldrich, United Kingdom) and 10% glycerol (Calbiochem, Germany). For infection, $1 \times 10^6$ *Pb*A- iRBC (100 µl of blood mixed with Alsever's solution) were injected intraperitoneally (i.p.) into each mouse. The infected mice were monitored daily from day 3 p.i. for clinical and neurological symptoms. The mice were evaluated based on the Rapid Murine Coma and Behaviour Scale (RMCBS)[71]. Mice were scored and divided into different groups based on their RMCBS: RMCBS score 15–20 (day 0–4 p.i.; asymptomatic), RMCBS score 10–14 (day 5–6 p.i.; early ECM) and RMCBS score 0–9 (day 7–8 p.i.; severe ECM). Parasitemia was measured daily after infection by microscopic examination of Diff-Quik (Medion Grifols Diagnostics, Switzerland) -stained thin blood smears.

### Anti-malarial treatment
Starting at day 5 p.i., one group of *Pb*A-infected mice was treated with the anti-malarial drug pyrimethamine (Sigma–Aldrich, Germany). 25 mg of pyrimethamine was dissolved in 1 ml of dimethyl sulfoxide (DMSO; Sigma–Aldrich, United Kingdom). 50 µl of this solution was dissolved with 950 µl of PBS (Gibco, United Kingdom) to make a working concentration of 1.25 mg/ml. Pyrimethamine was administered intravenously (i.v.) into mice (0.125 mg in 100 µl/mouse). The other group received phosphate-buffered saline (PBS) as control and has been referred to as untreated mice.

### Evans blue staining

Mice were i.v. injected with 100 µl of 2% Evans blue solution (Sigma–Aldrich, USA) prepared in PBS ($n = 2$ per group of mice). One hour later, the mice were anesthetized with 4.5–5.0% isoflurane in 2:1 $O_2$:$N_2O$ volume ratio and perfused transcardially with 0.9% NaCl (Carl Roth, Germany). The perfused brains were removed and photographed using a Sony Alpha A200K digital camera.

### Quantitative real-time-PCR (q-PCR)

mRNA was isolated from the brains of the indicated groups of mice using an RNAeasy kit (Qiagen, Germany) ($n = 4$ per group). The SuperScript reverse transcriptase kit with oligo (dT) primers (Invitrogen, Canada) was used to transcribe the mRNA into cDNA. Quantitative-PCR for interferon-γ (IFN-γ), tumor necrosis factor (TNF), chemokine (C-X-C motif) ligand 3 (CXCL3), CXCL9, CXCL10, CXCL11, lymphotoxin α (LTα), LTβ and hypoxanthine phosphoribosyltransferase (HPRT) was performed using cDNA from olfactory bulb, cortex, and brainstem regions, respectively, of uninfected (day 0) and infected mice (day 5 and 7 p.i.) with and without pyrimethamine-treated mice employing the respective Taqman gene expression assay (Applied Biosystems, Germany). To determine the parasite load in the different regions of the brain, q-PCR was performed for *P. berghei* ANKA cytochrome B (PBANKA_MIT01900)[72] and HPRT (Eurofins Genomics, Germany) using Light cycler 480 SyBr Green I master mix (Roche, Germany). Amplification was performed with a Light cycler 480 (Roche, Germany). Quantitation was performed using the Light cycler software SDS (version 5.1; Applied Biosystems, Germany), according to the $ddC_T$ threshold cycle method with HPRT as the housekeeping gene[73]. Data are depicted as the increase in the level of mRNA expression in infected mice over uninfected control mice.

### SPECT/CT imaging

For SPECT-imaging, all animals were implanted, under isoflurane anesthesia (1.5–3% isoflurane, 850 ml/min $O_2$), with catheters in the right external jugular vein[74] and were given at least one day to recover.

***SPECT-imaging of the biodistribution of $^{99m}$Tc-labeled iRBCs.*** Blood was collected from C57BL/6 *Pb*A-infected mice at day 7 p.i. with a parasite load of around 5% by cardiac puncture under isoflurane anesthesia. Enrichment of infected erythrocytes was performed by a Percoll gradient[75] with a yield of around 95% infected red blood cells. Around $7.5 \times 10^5$ infected erythrocytes were incubated with freshly prepared $^{99m}$Tc-HMPAO for 30 min. Solutions were then centrifuged for 5 min at 500 g, the supernatant removed, and the cells resuspended in 0.9% NaCl. Cell viability was controlled using trypan blue staining. Awake mice were i.v. injected over 6 min with the labeled iRBCs (24.1 MBq on average) in volumes of 300 µl using perfusion pumps. Dynamic whole-body scans of 15 min duration starting shortly after injection were used in pilot experiments to determine spatial redistributions after injection. As no evidence for redistribution was found, mice (C57BL/6 wt d0 $n = 7$, 3 female, 4 male, mean age 11 weeks; d5 p.i., $n = 8$, 3 female, 5 male mean age 10.6 weeks; C57BL/6 *prf*$^{-/-}$ d0 $n = 6$, male, mean age 16.9 weeks, d5 p.i $n = 6$, male, mean age 20.2 weeks; BALB/c d0 $n = 6$, male, mean age 9.5 weeks, d5 p.i $n = 6$, male, mean age 9.5 weeks) were imaged with a single whole-body scan with an acquisition time of 20 min followed by a single head scan with an acquisition time of 90 min. After injection of the labeled iRBCs mice were anesthetized and transferred to the scanner. Scanning started with whole-body SPECT/CT followed by head SPECT/CT. The head SPECT started one hour after injection of the labeled iRBCs.

SPECT/CT imaging was performed with a four-head NanoSPECT/CT™ scanner (Mediso, Hungary). Animals were scanned under gas anesthesia (1.2–1.5% isoflurane, 850 ml/min $O_2$). CT and SPECT were co-registered. Head scans were accompanied by two co-registered CT scans, one before and one after the SPECT scan, to control for motion

artefacts (none detected). CT scans were made at 45 kVp, 177 µA, with 180 projections, 500 ms per projection, and 96 µm isotropic spatial resolution, reconstructed with the manufacturer´s software (InVivoScope 1.43) at isotropic voxel-sizes of 200 µm for whole-body scans and 100 µm for head scans. SPECT scans were made using nine-pinhole mouse brain apertures with 1.2 mm pinhole diameters providing a nominal spatial resolution ≤1 mm. Photopeaks were set to the default values of the NanoSPECT/CT for $^{99m}$Tc (140 keV ± 5%). SPECT images were reconstructed using the manufacturer´s software (HiSPECT™, SCIVIS, Germany) at an isotropic voxel size of 448 µm in whole-body scans and 300 µm in head scans.

For analysis, whole-body CTs were used to create, in the OsiriX™ software, whole-body volume-of-interests (VOIs) for calculation of whole-body volumes and of $^{99m}$Tc-contents in the whole-body SPECT data. Whole-body $^{99m}$Tc-concentrations were calculated for each individual. All head SPECT data sets were aligned to a T2w anatomical reference MR. The alignment was based solely on the co-registered CT images. An automated workflow was used for the CT / MR alignment (see supplemental methods CT/SPECT registration). Briefly, the hyperintense skull was automatically segmented on CT images and the volume inside the skull was used to define a brain mask which was registered to the brain mask of the MRI. The calculated co-registration was then applied to the SPECT images.

For better analysis of the spatial relationship to the rostral rhinal vein all data sets were then aligned, using the same coordinate transformation, to a high-resolution MR dataset from the literature, the DSURQE-MR[76]. This dataset is from an ex vivo contrast-agent perfused mouse, in which the rrv is clearly visible. SPECT head data sets were then divided by the according whole-body concentrations to get data sets in units of standardized uptake values (SUV). SUV in whole-brain VOIs and VOIs on the confluence of the sinuses and rrv were measured in OsiriX™ (version 5.9.1, Pixmeo, Switzerland). Statistical testing using student *t*-test (unpaired, two-tailed, heteroscedastic) was done in Microsoft Excel™. Individual SPECT and CT head data sets of each group were added and divided by the number of animals per group in order to get group-mean data sets. Group-mean data sets and differences between group-mean SPECT data were overlaid on the anatomical reference MR and a group-mean MR-angiography from d0 mice in OsiriX™. Images from selected sections were arranged for illustrations in Photoshop™ (version 24, Adobe, USA). For clarity, extracranial parts of the SPECT data were removed in the images.

***SPECT-imaging of cerebral blood flow.*** Awake mice (d0 $n = 10$, male, mean age 10.5 weeks; d5 p.i. $n = 9$, male, mean age 9.8 weeks) during ongoing behavior were i.v. injected via the external jugular vein catheter with freshly prepared $^{99m}$Tc-HMPAO[74,77]. Doses of on average 102 MBq were injected in 330 µl within 10 min using a perfusion pump.

SPECT/CT head scans and alignment of the SPECT data to the anatomical reference brain were made as described above with an acquisition time of 1 h. After alignment, SPECT data were converted to TIFF-files, brain data were cut out from the head scans using a brain mask and global-mean normalized in ImageJ. An unpaired two-tailed heteroscedastic voxel-wise *t*-test was made in MATLAB (version R2017b (9.3.0.713579) 64-bit, Mathworks, USA). *T*-test result files (uncorrected for multiple comparisons) as well as group-mean and difference-files were converted to DICOM and overlaid on the anatomical reference MR in OsiriX™. Specific sections were chosen and the figures were edited and arranged in Photoshop™.

### MRI

Infected mice and controls underwent MR-imaging sessions including MR-TOF angiographies combined with T2w anatomical reference scans. Mice were scanned before infection, at day 5 p.i and at day 7 p.i. (C57BL/6 wt d0 $n = 5$, male, mean age 11 weeks, d5 $n = 10$, male, mean age 12.7 weeks, d7 $n = 10$, male, mean age 12.7 weeks; C57BL/6 *prf*$^{-/-}$

$n = 6$ male, mean age 15.8 weeks in each group; BALB/c $n = 5$, male, mean age 11 weeks in each group). Imaging of the brain was performed using a combination of a $^1$H volume coil of 86 mm in inner diameter for transmission and a $^1$H $3 \times 1$ surface array coil for receiving the signal. The MRI protocol included a MAPshim method for local shimming of the entire brain, a T2-weighted 2D TurboRARE (TR/TE = 4200 ms/ 18.5 ms, 2 averages) with equal field of view (25.6 × 25.6 mm) serving as anatomical reference for the equivalent MR-angiography. A non-contrast-enhanced 2D TOF with flow-compensation was acquired with optimized parameters (TR/TE: 3 ms/12 ms, in 80 slices with a slice thickness of 0.3 mm, slice overlaps of 0.1 mm and an in-plane-resolution of 0.08 × 0.08 mm resulting a scan time of ~4 min). The flip angle was set to 70° and the saturation slice, with a thickness of 3 mm, was positioned with a 1 mm gap towards the caudal side of the brain in order to image flow in caudo-rostral direction.

Anesthesia of the animals was induced with 3% Isoflurane carried by a 1:1 mixture of $O_2/N_2O$ followed by a reduction to 1% Isoflurane while MR-imaging was performed. Vital parameters of the mice, such as body temperature and breathing rate, were permanently monitored and kept constant during scanning procedure.

For analysis, T2w and TOF images were aligned to the same anatomical MR reference space (see supplemental methods TOF-image registration). Briefly, a group average TOF was registered to each T2 MRI and used as a reference to align individual TOF images with corresponding T2-MR-images. T2-MR-images were then registered to the Allen brain atlas, the co-registration was then applied to the TOF to yield an angiogram in reference space.

Average group angiography images were made in the same manner as described in the SPECT/CT experiments. The files were overlaid in OsiriX™ on MR T2 images of single individuals from each group. Images for the figures were arranged in Photoshop™ (version 24).

For calculating the signal intensities in the superior sagittal sinus (sss), a volume-of interest (VOI) file was created manually in OsiriX™ from all overlaid and aligned SSS in the control group. Mean signal intensities in the VOI in each individual were calculated using OsiriX™.

## Histology

The histological analysis of the large draining veins was done on 7 μm thick Giemsa-stained paraffin sections from material fixed by immersion in 4% paraformaldehyde in 100 mM phosphate buffer (PB) pH 7.4 and decalcified in Surgipath™ Decalcifier II (Leica, Germany). Mice were euthanized with isoflurane. The skin overlaying the dorsal cranium was removed. For better penetration of the fixative parts of the cranium were removed in parietal regions. In addition, a small whole of about 1 mm diameter was drilled into the frontal bones on each side. Care was taken to leave the bone intact above sinuses and the rostral rhinal vein. When removing bones, a distance of at least 1 mm to the sinuses was kept. Material from non-exsanguinated animals was used. After skull preparation the entire bodies were fixed in 4% PFA. The heads were removed from the bodies after fixation and the remaining skin was removed. All heads were stored in 1% PFA for no longer than five days. Heads were washed 3 × 10 min in PB, decalcified for 2 days, washed 3 × 10 min in 0.9% NaCl, dehydrated in an ascending ethanol series, and embedded in paraffin (Paraplast Plus, Roth, Germany). Seven micrometer thick whole-head sections were made on a sliding microtome (HM 400). Sections were mounted on gelatinized Superfrost® Plus glass slides that were covered with a drop of an albumin-glycerol solution before mounting. After drying sections were Giemsa-stained (Giemsa-solution, Roth, Germany) and covered with Merckoglas® and coverslips. Data from *Pb*A-infected C57BL/6 wt mice ($n = 3$ at d5, 2 female, 1 male, mean age 9 weeks; $n = 1$ at d6 p.i. female, 9 weeks; $n = 3$ at d7 p.i. 2 female, 1 male, mean age 9 weeks), C57BL/6 prf$^{-/-}$ mice ($n = 3$ at d7 p.i.,male, mean age 9.7 weeks) and *Pb*NK65-infected C57BL/6 wt mice ($n = 4$ at d7 p.i., male, mean age 8 weeks)

were analyzed microscopically. Photomicrographs were taken with a Histoscanner (3DHistech). High-resolution oil immersion images with and without polarizing filters were made on a Leica DMR microscope. The findings as displayed in Figs. 4, 5, and S2 were reproducible in each animal in the according groups. The anatomical relationships displayed in Fig. 6 were present in all mice studied in these groups.

## Immunohistochemistry (IHC)

For immunohistochemistry, mice were perfused transcardially with 0.9% NaCl followed by 4% paraformaldehyde (PFA; Carl Roth, Germany) solution in PBS ($n = 3$ per group and time point) after isoflurane anesthesia (4.5–5.0% isoflurane in 2:1 $O_2$:$N_2O$ volume ratio). The brains were removed, post-fixed in 4% PFA solution for 48 h, and cryoprotected in 30% sucrose (Sigma–Aldrich, USA) solution for 72 h followed by freezing in isopentane cooled by liquid nitrogen. Sagittal sections (50 μm) of frozen brain tissue were cut on a microtome (Leica CM 1950, Leica, Germany) and processed for hematoxylin (Mayer's Hematoxylin solution, Sigma–Aldrich, USA) and eosin (Merck, Germany) (H&E) staining. Immunostaining was performed with primary antibodies against glial fibrillary acidic protein (GFAP; rabbit, 1:10000, #z0334, Dako Denmark AS, Denmark), ionized calcium-binding adapter molecule 1 (Iba1; rabbit, 1:2000, #019-19741, Wako Chemicals GmbH, Germany), cleaved caspase-3 (Casp3; rabbit, 1:500, #9661 L, Cell Signalling Technology Europe B.V., Germany), CD31 (rat, 1:2000, #550274, BD Biosciences, Germany), and CD8 (rat, 1:1000, #14-0808-82, eBioscience, Affymetrix Inc., USA), followed by staining with the appropriate biotinylated secondary antibodies (anti-rabbit: 1:200, #111-065-144, Dianova, Germany and anti-rat: 1:200, #712-065-153, Dianova, Germany), and avidin-biotin-peroxidase complex (ABC kit, Vector Laboratories, USA). Diaminobenzidine (DAB; Sigma–Aldrich, USA) was used to visualize the antibody reactions. The sections were mounted on gelatine-coated slides and covered using coverslips and mounting solution, Merckoglas (Merck, Germany). Images of different brain regions including olfactory bulb (OB), sub-ventricular zone (SVZ), rostral migratory stream (RMS), hippocampus, midbrain and pons, were acquired using a digital camera (Leica DFC 500, Leica, Germany) mounted on a combined brightfield and epifluorescence microscope (Zeiss Axioskop 2, Zeiss, Germany).

For bromodeoxyuridine and doublecortin immunofluorescence (IF) staining bromodeoxyuridine (BrdU; 50 μg/g body weight; eBioscience, USA) was i.p. administered at day 3 and day 5 p.i. The mice were transcardially perfused 48 h post injection, ($n = 3$ per group), as aforementioned, with 0.9% NaCl and 4% PFA after isoflurane anesthesia. Brains were fixed in 4% PFA for 48 h followed by cryoprotection in 30% sucrose solution for 72 h and were frozen in liquid nitrogen-cooled isopentane. Sagittal sections (50 μm) were immunostained with primary antibodies against doublecortin (DCX; goat, 1:1000, #sc-8066, Santa Cruz Biotechnology, USA) followed by incubation with immunofluorescent Alexa-488 labeled secondary antibody (anti-goat, 1:200, #A11055, Invitrogen by Thermo Fischer Scientific, USA). The sections were subsequently incubated in 2 N HCl at 37 °C for 1 h to denature the genomic DNA followed by neutralization in 0.1 M borate buffer. Primary antibodies against BrdU (rat, 1:2000, #ab6326, Abcam, United Kingdom) were added and counterstained by immunofluorescent Cy3 labeled secondary antibody (anti-rat, 1:200, #712-165-153, Dianova, Germany). Sections were placed on gelatin-coated slides and covered with coverslips and mounting solution, MOWIOL (Fluka, Germany). The rostral migratory stream (RMS) was imaged with a digital camera (Leica DFC 500, Leica, Germany) mounted on the combined brightfield and epifluorescence microscope (Zeiss Axioskop 2, Zeiss, Germany).

For IHC analysis, every single sagittal brain section was photographed with a digital camera (Leica DFC 500, Leica, Germany) mounted on a combined brightfield and epifluorescence microscope (Zeiss Axioskop 2, Zeiss, Germany). For H&E staining, images were

acquired using 20× objective under brightfield illumination with exposure time of 8 ms. Cy3 conjugated BrdU antibodies were excited at 570 nm and Alexa-488 conjugated DCX antibodies were excited at 493 nm and images were captured using 5× objective in dark with exposure time of 590 ms and 650 ms, respectively. Images were acquired using a 20× objective under constant brightfield illumination for GFAP, Iba1, CD31, CD8 and cleaved Caspase-3 staining and their exposure times were 10 ms, 8 ms, 7 ms, 8 ms and 7 ms respectively. The respective images acquired were used for cell counting. Magnified images were captured under 40× objective for all the staining. For analysis and cell counting, ImageJ software (NIH, USA) was employed. The RGB images were converted to 16-bit gray scale images and threshold values were adjusted for each individual stain to highlight the structures to be counted. To reduce background noise in the images, background subtraction with rolling ball was carried out and particles were analyzed. With the exception of CD8 staining, cell number for all other staining were measured by automated counting the area covered by each cell type in the whole area of the image. For CD8 staining, the total number of CD8+ cells in the whole image were counted manually with the Cell Counter plugin in ImageJ.

### CT/SPECT registration

The image registration to standard space (Allen Mouse Brain atlas 2017 (CCFv3); http://help.brain-map.org/display/mouseconnectivity/API) was performed using the ANTx2 toolbox (https://github.com/ChariteExpMri/antx2) running in MATLAB (R2016b, Natick, Massachusetts, The MathWorks Inc.). Detailed information for registration can be found in Koch et al. [28]. For image registration to standard space (Fig. S12), the T2w image was rigidly registered to the Allen Brain template and segmented into the three tissue compartments (gray matter, white matter, and cerebrospinal fluid) using SPM's unified segmentation approach[78]. The tissue segments where further used for affine and subsequent nonlinear registration to the template tissue probability maps[79] using elastix[80] to obtain the T2w image in standard space.

The inverse transformation parameters based on the transformation parameters from rigid, affine, and nonlinear forward registration was used to back-transform the binary mask of the Allen mouse brain to the animal native space. The skull from the animal's CT image was extracted using Otsu and watershed segmentations, followed by image clustering and morphological image operations to finally obtain a binary CT-based brain mask. In the next step, the CT image and the SPECT image (in register with the CT image) were brought into to the animal's native space (T2w space). For this, the CT-based brain mask was rigidly registered to the Allen brain mask in native space and the same transformation parameters were the applied to the SPECT image. Finally, CT and SPECT images in native space were transformed to standard space by applying the forward transform parameters obtained from the first processing step. The CT/SPECT image registration workflow is implemented in the ANTx2 toolbox.

### TOF-image registration

The image registration to standard space (Allen Mouse Brain atlas 2017 (CCFv3); http://help.brain-map.org/display/mouseconnectivity/API) was performed using the ANTx2 toolbox (https://github.com/ChariteExpMri/antx2) running in MATLAB (R2016b, Natick, Massachusetts, The MathWorks Inc.). Detailed information for registration can be found in Koch et al. [28]. We first generated a TOF template by manual registration of TOF images from a group of animals to an individual T2w image using the MPI-Tool™ software (version 6.36, ATV, Advanced Tomo Vision, D-50169 Kerpen, Germany). For registration of the TOF template to standard space, the T2-weighted (T2w) image was rigidly registered to the Allen brain template, and segmented into the three tissue compartments (gray matter, white matter,

and cerebrospinal fluid) using SPM's (https://www.fil.ion.ucl.ac.uk/spm/) unified segmentation approach[78]. The tissue segments where further used for affine and subsequent nonlinear registration to the template tissue probability maps[79] using elastix[72] to obtain the T2w image in standard space. Using the concatenated transformation parameters from rigid, affine, and nonlinear registrations, the TOF template was finally transformed to standard space. For each animal, the T2w image from native space was transformed to standard space as described above. In a next step, the TOF template in standard space was back-transformed to each animal's native space using the inverse transformation parameters for registration to standard space. Next, the animal's TOF-image was rigidly registered to the TOF template in the animal's native space. Finally, the registered TOF-image was transformed from native space to standard space using the animal's concatenated transformation parameters (rigid, affine, and nonlinear registrations).

For each animal, the T2w image from native space was transformed to standard space as described above. In a next step, the TOF template in standard space was back-transformed to each animal's native space using the inverse transformation parameters for registration to standard space. Next, the animal's TOF-image was rigidly registered to the TOF template in the animal's native space. Finally, the registered TOF-image was transformed from native space to standard space using the animal's concatenated transformation parameters (rigid, affine, and nonlinear registrations).

### Statistics

Statistical analyses of percentage of appearance of symptoms, parasitemia, immunohistochemistry, and q-PCR, respectively, were performed using GraphPad Prism versions 9 and 10 (GraphPad, USA). Statistical differences between groups were determined using two-way ANOVA. All experiments were performed at least twice. $P$-values < 0.05 were considered to be * significant, $P$-values < 0.01 were ** significant, and $P$-values < 0.001 were *** significant.

### Reporting summary

Further information on research design is available in the Nature Portfolio Reporting Summary linked to this article.

## Data availability

All primary data associated with this study are available in the article and the supplementary information. The source data are provided as a Source Data file at: https://doi.org/10.6084/m9.figshare.25225802

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

## Acknowledgements

This project has been funded by the federal state of Saxony-Anhalt, Germany, and the European Structural and Investment Funds (ESF, 2014-2020), project number ZS/2016/08/80645 to G.N., D.S., E.B., and J.G. and the Deutsche Forschungsgemeinschaft (SFB 854, TP30 to G.N.). Funding to S.P.K. and P.B.S. was provided by the German Federal Ministry of Education and Research (BMBF) under the ERA-NET NEURON scheme (01EW1811 and 01EW2305), and the German Research Foundation (DFG, Project BO 4484/2-1 and EXC-2049-390688087 NeuroCure). Analysis of imaging data was supported by Charité 3R | Replace, Reduce, Refine. The authors wish to thank Holger Reim, Janet Stallmann, Anja Gürke, Marko Dombach, Gordon Kühne, Torsten Stoeter (Leibniz Institute for Neurobiology, Magdeburg), Annette Sohnekind, Nadja Schlüter, Anita Marquardt (Institute for Medical Microbiology, University Hospital, Magdeburg) for their excellent support and technical assistance in the experiments.

## Author contributions

G.N. and J.G. conceptualized and supervised the experiments. A.M.O., R.B., P.W., K.H., H.R., S.K., P.B., J.G. and G.N. performed experiments and analyzed data. A.M.O., R.B., P.W., K.H., K.M., E.B., D.S., J.G. and G.N. interpreted the data. G.N. and J.G. wrote the manuscript. D.S. and K.M. reviewed and edited the manuscript.

## Funding

## Competing interests

The authors declare no competing interests.

## Additional information

[1]Combinatorial NeuroImaging Core Facility, Leibniz Institute for Neurobiology, 39118 Magdeburg, Germany. [2]Research group Neuroplasticity, Leibniz Institute for Neurobiology, 39118 Magdeburg, Germany. [3]Institute of Medical Microbiology and Hospital Epidemiology, Hannover Medical School, 30625 Hannover, Germany. [4]Institute of Anatomy, Medical Faculty, Otto-von-Guericke-University Magdeburg, Leipziger Strasse 44, 39120 Magdeburg, Germany. [5]Charité-Universitätsmedizin Berlin, corporate member of Freie Universität Berlin and Humboldt-Universität zu Berlin, Department of Experimental Neurology and Center for Stroke Research, Charitéplatz 1, 10117 Berlin, Germany. [6]Charité-Universitätsmedizin Berlin, NeuroCure Cluster of Excellence and Charité Core Facility 7T Experimental MRIs, 10117 Berlin, Germany. [7]Charité-Universitätsmedizin Berlin, Charité 3R | Replace, Reduce, Refine, Charitéplatz 1, 10117 Berlin, Germany. [8]Department of Molecular Parasitology, Institute of Biology, Humboldt University, 10115 Berlin, Germany. [9]Center of Behavioural Brain Sciences, Universitätsplatz 2, 39106 Magdeburg, Germany. [10]These authors contributed equally: A. M. Oelschlegel, R. Bhattacharjee, P. Wenk. ✉e-mail: Juergen.Goldschmidt@lin-magdeburg.de; GopalaKrishna.Nishanth@mh-hannover.de

