## [Peer Review File · Nature Communications]

Beyond the microcirculation: sequestration of infected red blood cells and reduced flow in large draining veins in experimental cerebral malariaReviewers' Comments:

Reviewer #1:

Remarks to the Author:

Bhattacharjee et al. have studied the role of parasite sequestration on blood circulation during experimental cerebral malaria. Using various imaging technologies to track RBC and measure blood flow, they have demonstrated temporal parallelism of iRBC sequestration and reduction in global venous efflux out of the brain of infected mice. Although the study is mostly well designed, performed, and analyzed, it is mainly incremental of what we know for CM pathogenesis, that sequestration is essential for both CM and ECM to occur.

Additional comments

1. Line 53: add a reference (MacPherson et al, Am J Pathol, 1985, Silamut et al, Am J Pathol 1999)
2. Line 54: this is not completely true: in ECM, parasites have been seen in the other side of the BBB, although maybe not deep in the parenchyma (Strangward et al, PLoS Pathog, 2017). So, the authors should rephrase this claim and add the relevant references.
3. Line 59. The reference to experimental cerebral seems misplaced. There is no mention of this model before in the introduction. It should be moved to the ECM section.
4. Line 59: the reference that sequestration is debated is from 2007. A lot of work has been done since then showing a strong association of *P. berghei* sequestration with ECM occurrence and other pathologies (to cite a few: Amante et al, J. Immunol, 2010; Claser et al, PloS One, 2011; Howland et al, EMBO Mol Med, 2013; Khoury et al, Infect Immun, 2014; Matz et al, Eukaryot Cell, 2015; Possemier et al, PLoS Pathog, 2021, etc.)
5. Using antimalarials to prevent ECM is already standard. From a clinical point of view, adjunct therapy at the time of the CM symptoms is what is really needed.
6. Line 235: please cite all the references which demonstrated this in a more systematic manner (Amante et al, J. Immunol, 2010; Claser et al, PloS One, 2011).
7. Line 293: retinopathy is not always a marker of CM (Small et al, eLife, 2017; Shabani et al, BMC Med, 2017).
8. Line 476: there are many uncloned and cloned lines of PbA, which some demonstrate different levels of ECM-inducing capacity. The authors should provide more information on the origin of the line used in this study. Is it a clone line? Where does it originate from, etc.?

Reviewer #2:

Remarks to the Author:

This study uses elegant in vivo imaging to provide evidence for parasitised red cell accumulation and reductions in blood flow occurring in the larger veins and sinuses of the cerebral circulation during the development of experimental cerebral malaria in mice. The study provides novel data which will be of interest to those working in the field, however it falls short of demonstrating that the accumulation of parasites and decreased flow in the draining veins and sinuses play a causal role in the pathogenesis of experimental cerebral malaria. The relevance to human cerebral malaria is also uncertain.

The manuscript is generally well written and clearly reports and explains all procedures. However, presentation of results in figures using bars with unspecified error bars is not acceptable and individual data points would need to be shown along with an appropriate measure of central tendency and a specified measure of dispersion appropriate for the distribution of the data. The statistical analysis repeatedly uses student's t-test instead of more appropriate methods for comparison of multiple groups (for example ANOVA) and because the distribution of individual data points is not shown, it is not possible to determine whether parametric or non-parametric tests are more appropriate.

Major limitations of the study

Although the imaging findings are impressive, they are not accompanied by histological analysis of the

large veins and sinuses to confirm that there is genuinely sequestration of infected red blood cells causing obstruction of blood flow. This is curious given the amount of other histological analysis that has been conducted, and I would urge the authors to provide direct evidence that parasites do indeed sequester in these locations in sufficient quantities to impede blood flow, to corroborate the imaging findings. This is especially important because other authors have reported that little parasite accumulation occurs in larger blood vessels (Stangward et al 2017, <https://doi.org/10.1371/journal.ppat.1006267>). Curiously, this paper by Strangward et al., which is the most comprehensive histological and anatomical study of parasite sequestration in ECM to date, has not been referenced in the current paper.

This study does not establish whether the changes in venous flow play any causal role in the onset of ECM. The experiment with pyrimethamine treatment only shows that if parasites are killed before the onset of ECM, then mice do not develop ECM. This is unsurprising, and does not demonstrate any causal role for the vascular findings described. It would be instructive to investigate whether the same vascular findings are seen in *P. berghei* NK65 infection (which does not cause ECM in C57Bl/6 mice), in *Pb ANKA* infection of BALB/c mice (which are resistant to ECM), and particularly in perforin deficient mice infected with *Pb ANKA* (which do not develop ECM). It would also be ideal to establish whether preventing sequestration of iRBCs in the veins during *PbANKA* infection of wild type mice can prevent the onset of ECM, although this would be challenging because the mechanisms of cytoadhesive sequestration by *Pb ANKA* remain unclear.

The relevance to human disease is dubious, because it seems far less likely that iRBC accumulation will occur to a sufficient extent to obstruct or significantly impair the much larger veins and venous sinuses of humans, and indeed there is little evidence from post-mortem studies that this does occur. The authors point out that moderate narrowing of the lumen by iRBCs in mice would impact flow because of the Poiseuille law where resistance to flow is proportional to the fourth power of radius. However, the relative reduction in radius of a much larger human vessel would have relatively much smaller impact on flow. The issue of differences in the size of the microvasculature and the impact this has on obstruction by parasitised red cells was discussed in detail in the paper by Strangward et al 2017.

Other major comments

1. The developmental stage of the parasites in the iRBCs labelled with ^{99m}Tc need to be shown. Other work shows that microvascular sequestration occurs predominantly with mature parasites, so if early stage parasites are obtained from the circulation they are less likely to accumulate in the microvasculature and therefore may appear to accumulate in other locations.
2. All mention that the results demonstrate that iRBC accumulation is essential for ECM and particularly the pyrimethamine experiment demonstrating this, need to be removed unless direct evidence is provided. The pyrimethamine experiment only shows that ongoing infection is required for ECM, which is self-evident.

Other comments and clarifications required

1. Line 45: I do not believe that cerebral malaria accounts for 90% of malaria deaths globally and cannot find a reference to support this statement. It is not stated in the 2021 WHO World Malaria Report
2. Line 56: "discussed controversially" – rephrase
3. Line 56: "Main" – rephrase
4. Introduction - the current text does not do justice to the extensive debate in the literature about the relevance of ECM to human CM
5. Results: why has neuroimaging not been done in the pyrimethamine treated mice, to show whether all changes in blood flow rapidly resolve?
6. Materials and methods
 - a. The age and sex of the mice in each experiment needs to be reported
 - b. The source of the *PbANKA* parasites needs to be reported

- c. What is the effect of the lifecycle stage of the parasite on the method used to measure parasite load in the brain? Schizonts will contain more copies per iRBC of each parasite gene, therefore the qPCR measurement will be confounded by differences in parasite stage. Since more mature parasites sequester, it is highly likely that this is inflating estimates of the number of infected RBCs in the brain.
- d. What day of infection and at what parasitemia were RBCs collected for 99mTc labelling?
- e. Line 535 – cells are resuspended not dissolved
- f. To what extent might anaesthesia and the invasive procedures necessary for some of the imaging contribute to changes in blood flow in mice which are already unwell? It would be ideal to have another control group of mice with malaria infection (eg PbNK65) which are similarly unwell but do not have ECM, to show that changes in the cerebral circulation are specific to ECM

Reviewer #4:

Remarks to the Author:

This is a fascinating and well-written manuscript describing a logical series of experiments which establish the sequence of events culminating in fatal murine cerebral malaria (experimental cerebral malaria) in the *Plasmodium berghei* ANKA/C57BL/6 mouse model.

Using a combination of live imaging of parasite-infected red cells, brain imaging, post mortem histology, quantitative PCR of postmortem tissues, the investigators describe the relationship of the spatial distribution of iRBCs, impaired perfusion in areas of draining of large veins, impaired venous efflux from the brain in general, brain volume and clinical outcome. The piece-de-resistance is the demonstration of the reversal of many of these effects when the mice are treated with an antimalarial drug.

This is an excellent example of how a model system can be used to illuminate the pathogenesis of an important human disease. It is simply not possible to observe the temporal sequence of events in the course of the development of cerebral malaria in the human host.

The important findings, which significantly illuminate our understanding of disease pathogenesis in human CM, are that (a) iRBC accumulation and impaired perfusion precede the inflammatory response, (b) iRBCs accumulate in the microcirculation and in larger venous vessels, (c) impaired venous efflux leads to impaired perfusion in venous drainage territories (d) these are the first contributors to increased brain volume, but the ensuing inflammatory response exacerbates the brain swelling, and (e) an antimalarial intervention at the stage of impaired perfusion can prevent further increase in brain volume and the onset of the inflammatory response.

The striking similarities to human CM include:

- Involvement of venous watersheds. Many of the radiologic findings in CM survivors are transient --- they resemble strokes initially, but resolve --- suggesting that they are 'venous strokes' rather than arterial strokes (Seydel KB et al., NEJM 2015).
- Although sequestration of iRBCs occurs in gray and white matter in human CM, ring hemorrhages are largely restricted to cortical white matter (Milner D, et al., 2014). These investigators observed the same phenomenon, and attributed it to this area being a venous watershed.
- The rapid reversibility of many radiologic features after treatment (Seydel KB et al., NEJM 2015)
- Similarity of pathology in individuals with "sequestration only" (in humans) (Taylor TE, Nat Med, 2004) and 'impaired perfusion only' (in mice).

The authors are careful to describe 'parasite accumulation' in their model, and largely but not always (eg, line 232) distinguish it from 'parasite sequestration' in human CM.

Parasite sequestration involves specific ligands on the surface of the parasitized red cell, and specific receptors on the surface of endothelial cells. Those interactions are thought to evoke the inflammatory response in human CM.

Q1: Is there any evidence of sequestration in this murine model?

Q2: Are the authors referring to 'parasite accumulation' because their methods lacked sufficient resolution to identify sequestration, or is the phenomenon truly a preferential accumulation of iRBCs in the lumen of larger venous vessels?

Q3: Absent a specific interaction with endothelial cells, can the authors propose a mechanism for the accumulation of iRBCs? Could it be related to the decreased O₂ tension in venous blood?

Q4: Can the authors suggest a mechanism by which accumulations of iRBCs, in the absence of an interaction with endothelial cells, would result in an inflammatory response? Could there be a rheologic effect (e.g., Herricks T, et al., Cell Microbiol 2012)? Could it be hydrostatic?

Q4: In human CM, there is abundant evidence of iRBCs in the cerebral microcirculation. Venous vessels which would normally empty after death are often distended by iRBCs. This is consistent with the authors' observations of increased brain volume in association with the accumulations of iRBCs. It is unfortunate that the methods used in this manuscript were not sensitive enough to observe individual vessels. Absent that, it is difficult to determine if the decreased venous efflux originates in smaller, intracerebral vessels and the ensuing decreased flow results in accumulations of iRBCs in larger veins or if the accumulations of iRBCs in the larger draining veins and sinuses are the inciting events, and, in effect, 'dam up' flow from the upstream veins and post-capillary venules. Can the authors illuminate this 'chicken and egg' question? Which came first, obstruction of small vessels or the accumulation of iRBCs in larger draining veins and sinuses?

The temporal relationship between the administration of pyrimethamine and the abrogation of the inflammatory response is striking and is one of the real strengths of the paper.

Q5: What was the rationale for choosing pyrimethamine? It is an older antimalarial, and is rarely, if ever, used now. There are newer, faster drugs. The choice of this antimalarial should be justified by the authors.

Overall, this manuscript, by highlighting the potential importance of impaired venous efflux and the reversibility of the early changes associated with it, opens up important new areas of study related to CM pathogenesis and treatment. Establishing the temporal association between impaired venous drainage and subsequent pathophysiologically significant events (at least in the murine model) really changes the landscape and helps to explain earlier observations.

Terrie E. Taylor, D.O.

Response to reviewer comments: We thank the reviews for critically reading the manuscript and providing constructive feedback. We have addressed all the points raised by the reviewers by performing additional experiments and providing further clarifications.

Reviewer #1:

Bhattacharjee et al. have studied the role of parasite sequestration on blood circulation during experimental cerebral malaria. Using various imaging technologies to track RBC and measure blood flow, they have demonstrated temporal parallelism of iRBC sequestration and reduction in global venous efflux out of the brain of infected mice. Although the study is mostly well designed, performed, and analyzed, it is mainly incremental of what we know for CM pathogenesis, that sequestration is essential for both CM and ECM to occur.

Response: We thank the reviewer for the appraisal of our study. We wish to highlight that while sequestration of iRBCs in the microcirculation is the main contributing factor in human CM, it remains controversial in ECM. Most importantly, the novelty of this study is that iRBCs accumulate not only in the microcirculation but also in large draining veins and sinuses, particularly in the superior sagittal sinus and the rostral rhinal vein, which constitute the main route of venous efflux in mice. This finding may also prompt reconsidering the role of venous efflux reductions in large draining veins in human cerebral malaria.

Additional comments

1. Line 53: add a reference (MacPherson et al, Am J Pathol,1985, Silamut et al, Am J Pathol1999)

Response: The suggested references are now included (page 3, Line 60, reference numbers 7 and 8).

2. Line 54: this is not completely true: in ECM, parasites have been seen in the other side of the BBB, although maybe not deep in the parenchyma (Strangward et al, PLoS Pathog, 2017). So, the authors should rephrase this claim and add the relevant references.

Response: The reviewer raises an important point. The sentence has now been rephrased and reads as follows (page 3, lines 60-63):

“In contrast to other parasitic diseases of the CNS, *Pf*-infected red blood cells (iRBCs) in CM remain primarily intravascular. However, in murine experimental CM (ECM) some iRBCs have also been observed perivascularly in the parenchyma”

The reference has been included as reference number 14, on page 3 line 63

3. Line 59. The reference to experimental cerebral seems misplaced. There is no mention of this model before in the introduction. It should be moved to the ECM section.

Response: The misplaced reference has been removed.

4. Line 59: the reference that sequestration is debated is from 2007. A lot of work has been done since then showing a strong association of *P. berghei* sequestration with ECM occurrence and other pathologies (to cite a few: Amante et al, J. Immunol, 2010; Claser et al, PloS One, 2011; Howland et al, EMBO Mol Med, 2013; Khoury et al, Infect Immun, 2014; Matz et al, Eukaryot Cell, 2015; Possemier et al, PLoS Pathog, 2021, etc..)

Response: As suggested, we have incorporated more recent references into the text (page3, line 72, reference numbers 19-24).

5. Using antimalarials to prevent ECM is already standard. From a clinical point of view, adjunct therapy at the time of the CM symptoms is what is really needed.

Response: We completely agree with the reviewer. However, the rationale behind using anti-malarial treatment was to dissect the chronology of the events that contribute to ECM pathology i.e. iRBC sequestration and the pro-inflammatory response. Our data show that i) early iRBC-dependent brain pathology is present as early as day 5 p.i., when no clinical signs are visible, ii) that progression of brain pathology and clinical symptoms depend on iRBCs (pyrimethamine experiment) and iii) that without contribution of perforin-producing immune cells brain pathology, including iRBC sequestration and accumulation, and clinical symptoms do not progress (new experiments with perforin-deficient mice).

6. Line 235: please cite all the references which demonstrated this in a more systematic manner (Amante et al, J. Immunol, 2010; Claser et al, PloS One, 2011).

Response: As suggested, the references have now been included (page 3, lines 72, reference numbers 19 and 20).

7. Line 293: retinopathy is not always a marker of CM (Small et al, eLife, 2017; Shabani et al, BMC Med, 2017).

Response: We thank the reviewer for raising this important point and have rephrased the sentence, which now reads (Page 17, lines 404-406):

“Several studies have shown that malaria retinopathies can indicate CM in infected neurologically symptomatic patients”

8. Line 476: there are many uncloned and cloned lines of PbA, which some demonstrate different levels of ECM-inducing capacity. The authors should provide more information on the origin of the line used in this study. Is it a clone line? Where does it originate from, etc.?

Response: We used in our study the 'cl15cy1' of the ANKA strain of *P. berghei*. This information is now included (page 29, line 694 reference number 71).

Reviewer #2 (Remarks to the Author):

This study uses elegant *in vivo* imaging to provide evidence for parasitised red cell accumulation and reductions in blood flow occurring in the larger veins and sinuses of the cerebral circulation during the development of experimental cerebral malaria in mice. The study provides novel data which will be of interest to those working in the field, however it falls short of demonstrating that the accumulation of parasites and decreased flow in the draining veins and sinuses play a causal role in the pathogenesis of experimental cerebral malaria. The relevance to human cerebral malaria is also uncertain.

Response: We thank the reviewer for the appraisal of our experimental approach. We also appreciate the notion that the impact of parasite accumulation and decreased flow in the draining veins and sinuses on ECM pathogenesis should be further substantiated. We have generated a full range of new histological data sets to distinguish iRBC accumulation *vs.* sequestration and generated more *in vivo* data to corroborate our conclusion that reduced venous efflux plays a role in the onset of experimental cerebral malaria (ECM). We now provide further experimental support for our interpretation that reduced flow in the large draining veins largely influences parasite distribution and the edema. Experimentally, it is currently very difficult to design and show a definite proof for causation. We think it is of high relevance for future studies in the field to make clear that the pathological key events identified in our study could contribute to a better explanation to brain pathology in CM and provide new insights beyond the traditional microcirculation-centered theories.

The statistical analysis repeatedly uses student's t-test instead of more appropriate methods for comparison of multiple groups (for example ANOVA) and because the distribution of individual

data points is not shown, it is not possible to determine whether parametric or non-parametric tests are more appropriate.

Response: We agree with the reviewer and have now used ANOVA for the comparison of multiple groups wherever applicable (Fig 8, Fig 9, Fig S3 and Figure S10).

Major limitations of the study

Although the imaging findings are impressive, they are not accompanied by histological analysis of the large veins and sinuses to confirm that there is genuinely sequestration of infected red blood cells causing obstruction of blood flow. This is curious given the amount of other histological analysis that has been conducted, and I would urge the authors to provide direct evidence that parasites do indeed sequester in these locations in sufficient quantities to impede blood flow, to corroborate the imaging findings. This is especially important because other authors have reported that little parasite accumulation occurs in larger blood vessels (Stangward et al 2017, <https://doi.org/10.1371/journal.ppat.1006267>). Curiously, this paper by Stangward et al., which is the most comprehensive histological and anatomical study of parasite sequestration in ECM to date, has not been referenced in the current paper.

Response: We apologize for the oversight and have now cited the study of Stangward et al. (ref 14). Most importantly, we have now included histological analyses showing iRBCs distributions corroborating our results from *in vivo* imaging. The new results are displayed in figures 4-6 and S2.

This study does not establish whether the changes in venous flow play any causal role in the onset of ECM. The experiment with pyrimethamine treatment only shows that if parasites are killed before the onset of ECM, then mice do not develop ECM. This is unsurprising, and does not demonstrate any causal role for the vascular findings described.

Response: The rationale behind using anti-malarial treatment was to dissect the chronology of the events that contribute to ECM pathology i.e. iRBC sequestration and the pro-inflammatory response. Our data show that i) early iRBC-dependent brain pathology is present as early as day 5 p.i., when no clinical signs are visible, ii) that progression of brain pathology and clinical symptoms depend on iRBCs (pyrimethamine experiment) and iii) that without contribution of perforin-producing immune cells brain pathology, including iRBC sequestration and accumulation, and clinical symptoms do not progress (new experiments with perforin-deficient mice).

It would be instructive to investigate whether the same vascular findings are seen in P berghei NK65 infection (which does not cause ECM in C57Bl/6 mice), in Pb ANKA infection of BALB/c mice (which are resistant to ECM), and particularly in perforin deficient mice infected with Pb ANKA (which do not develop ECM). It would also be ideal to establish whether preventing sequestration of iRBCs in the veins during PbANKA infection of wild type mice can prevent the onset of ECM, although this would be challenging because the mechanisms of cytoadhesive sequestration by Pb ANKA remain unclear.

Response: The reviewer suggests a number of control experiments employing infection models that do not lead to ECM and permit a definitive distinction between signatures of infection dynamics and early pathological events that precede ECM and signify progression to edema. While the infections are straightforward the advanced imaging techniques and data analysis, which now comprises >200 *in vivo* imaging data sets, posed a substantial work load to this project. Yet, we agree that the recommended experiments further strengthen the novel findings of our study. As suggested, we performed additional experiments addressing factors driving iRBC accumulation in veins, including the role of perforin for RBC accumulation. Accordingly, we performed a range of new infection experiments with *PbANKA* parasites in perforin-deficient mice, which we analyzed with iRBC-SPECT, MR imaging and by histology. Additionally, as

per reviewer's request, we analyzed i) BALB/c infected with *PbANKA* by *in vivo* imaging (iRBC-SPECT and MR) and ii) *PbNK65*-infected C57BL/6 histologically. These studies show that, in contrast to the massive reduction in flow monitored in *PbANKA*-infected C57BL/6 mice, which later develop edema, there was no reduction in flow in *PbANKA*-infected BALB/c mice, and only a minor reduction of flow in C57BL/6 *prf*^{-/-} mice. Thus, these control experiments, as proposed by the reviewer, further substantiate our discovery that reduced flow in the large veins is both a hallmark and the earliest sign of an infection that is on a ECM trajectory. We are very satisfied with the experimental outcomes and hope the reviewer agrees with our assessment.

The relevance to human disease is dubious, because it seems far less likely that iRBC accumulation will occur to a sufficient extent to obstruct or significantly impair the much larger veins and venous sinuses of humans, and indeed there is little evidence from post-mortem studies that this does occur. The authors point out that moderate narrowing of the lumen by iRBCs in mice would impact flow because of the Poiseuille law where resistance to flow is proportional to the fourth power of radius. However, the relative reduction in radius of a much larger human vessel would have relatively much smaller impact on flow. The issue of differences in the size of the microvasculature and the impact this has on obstruction by parasitised red cells was discussed in detail in the paper by Strangward et al 2017.

Response: Again, we see the important point raised by the reviewer and agree with respect to potentially small effects of sequestration *per se* on the diameter of a large vessel. Accordingly, we removed the reference to the "Poiseuille" law.

The relevance of our study for the human disease is, in our opinion, high, because we present, for the first time, data on reduced efflux through the large veins and sinuses that offer a plausible explanation for subsequent edema formation and the spatial distribution of haemorrhages.

While potential effects of sequestered iRBCs on large vessel diameters may be less pronounced, the overall impact on flow could nevertheless be serious, in particular in combination with additional intravascular effects, like rosetting and altered deformability. We propose that critical consideration whether this early pathological signature could explain the edema dynamics is warranted. The assumption of spatiotemporal sequences of single-capillary obstructions or inflammation is challenged by our data, and we consider this an important and truly novel contribution to the field. In this regard our experimental murine data might stimulate imaging studies addressing blood flow in larger veins and sinuses in human CM patients.

Other major comments

1. The developmental stage of the parasites in the iRBCs labelled with ^{99m}Tc need to be shown. Other work shows that microvascular sequestration occurs predominantly with mature parasites, so if early stage parasites are obtained from the circulation they are less likely to accumulate in the microvasculature and therefore may appear to accumulate in other locations.

Response: We agree with the reviewer and have now checked for the parasite stage in the RBCs used for labelling. They contain almost exclusively mature forms, *i.e.* trophozoites and schizonts, which are known to sequester at the brain endothelium, while ring stages, which still circulate in the blood and do not yet sequester, were virtually absent. A representative example of the iRBCs prior to labeling is shown below.

2. All mention that the results demonstrate that iRBC accumulation is essential for ECM and particularly the pyrimethamine experiment demonstrating this, need to be removed unless direct evidence is provided. The pyrimethamine experiment only shows that ongoing infection is required for ECM, which is self-evident.

Response: As mentioned earlier, the rationale behind using anti-malarial treatment was to dissect the chronology of the events that contribute to ECM pathology i.e iRBC sequestration and the pro-inflammatory response. Pyrimethamine treatment was initiated at day 5, when functional impairment of blood flow was already detectable but cerebral bleeding, i.e. severe BBB damage, and cerebral inflammation were absent and mice were clinically completely healthy. Of note, *PbANKA*-specific CD8 T cells start to expand at this time, but only accumulate in the brain beyond day 5 p.i. Thus, the pyrimethamine experiments enabled us to show that iRBC at day 5 induce early brain pathology independent of intracerebral CD8+ T cell accumulation and that progression of ECM and brain pathology requires both iRBCs (pyrimethamine experiment) and pathogen-specific CD8+ T cells (perforin-deficient experiments).

Other comments and clarifications required

1. Line 45: I do not believe that cerebral malaria accounts for 90% of malaria deaths globally and cannot find a reference to support this statement. It is not stated in the 2021 WHO World Malaria Report

Response: We apologize for this error, which has now been rectified. The sentence now reads “Globally, an estimated number of 608,000 patients died of malaria in 2022, while 96 % of the malaria-related deaths occurred in sub-Saharan Africa, primarily in infants and children below the age of 5 years.”

2. Line 56: “discussed controversially” – rephrase

Response: This has been rephrased with the word “debated”.

3. Line 56: “Main” – rephrase

Response: This word “main” has been removed.

4. Introduction - the current text does not do justice to the extensive debate in the literature about the relevance of ECM to human CM

Response: We have elaborated on this controversy in the text now.

5. Results: why has neuroimaging not been done in the pyrimethamine treated mice, to show whether all changes in blood flow rapidly resolve?

Response: Pyrimethamine treatment was used to dissect the order of the events that contribute to ECM pathology *i.e.* direct effects of iRBC sequestration *vs.* down-stream effects the pro-inflammatory response. The reviewer's suggestion to comprehensively image the resolution of infection after therapeutic intervention is well appreciated. We will certainly consider this approach for future studies, since potential reduction of sequelae by adjunct therapy is an important avenue to follow.

6. Materials and methods

a. The age and sex of the mice in each experiment needs to be reported

Response: We now added this data.

b. The source of the PbANKA parasites needs to be reported

Response: This information has now been included. The reference line 'cl15cy1' of the ANKA strain of *P. berghei* was used for the infection now included on page 29 line 694 reference no 71.

c. What is the effect of the lifecycle stage of the parasite on the method used to measure parasite load in the brain? Schizonts will contain more copies per iRBC of each parasite gene, therefore the qPCR measurement will be confounded by differences in parasite stage. Since more mature parasites sequester, it is highly likely that this is inflating estimates of the number of infected RBCs in the brain.

Response: The reviewer raises a valid point. Therefore, we have now included histological stainings to show lifecycle stage of the parasite. We observe mainly the mature forms, *i.e.* trophozoites and schizonts sequestering at the brain endothelium.

d. What day of infection and at what parasitemia were RBCs collected for ^{99m}Tc labelling?

Response: The iRBCs were collected at day 5 p.i with a mean parasitaemia of 5%.

e. Line 535 – cells are resuspended, not dissolved

Response: The error has been rectified

f. To what extent might anaesthesia and the invasive procedures necessary for some of the imaging contribute to changes in blood flow in mice which are already unwell? It would be ideal to have another control group of mice with malaria infection (eg PbNK65) which are similarly unwell but do not have ECM, to show that changes in the cerebral circulation are specific to ECM

Response: Mice were anesthetized during the measurements inside the scanners (SPECT and MR), but in SPECT-imaging the labelled iRBCs and the blood flow tracer were injected in the awake state, and the images reflect the uptake patterns in the awake state (for details see our review in ref. 75). The TOF-MRA are from anesthetized mice, but we have now included further models as controls. We also refer to the study quoted in the MS (Rodriguez-Munoz, D., *et al.* ref. 37) showing, with TOF-MRA in hypothyroid PbA-infected mice, intact flow despite BBB-disruption.

Reviewer #4 (Remarks to the Author):

Q1: Is there any evidence of sequestration in this murine model?

Response: We can now show that iRBCs are in direct contact to the vessel walls. The same types of deposits that we detected in these iRBCs, i.e. dark-brown granules that polarized light, were also found in parts of the vessel walls suggesting vessel reactions and sequestration of iRBCs. These data are now included in figures 5 and 6.

Q2: Are the authors referring to ‘parasite accumulation’ because their methods lacked sufficient resolution to identify sequestration, or is the phenomenon truly a preferential accumulation of iRBCs in the lumen of larger venous vessels?

Response: Our new histological data suggest that there is true sequestration. But at the same time they also show that there are dense aggregations of iRBCs within the large draining veins. We argue that sequestration alone would likely not be sufficient for inducing severe reductions in flow, but rheopathological factors in conjunction with sequestration likely could. In short, it appears to be both, sequestration and accumulation.

Q3: Absent a specific interaction with endothelial cells, can the authors propose a mechanism for the accumulation of iRBCs? Could it be related to the decreased O₂ tension in venous blood?

Response: We can now show the interaction of iRBCs with the vessel walls and the transfer of iRBC material (most likely haemozoin) to the endothelial cell which is indicative of sequestration. However as the reviewer suggests the decreased O₂ tension in venous blood may certainly play a role. In addition, the region around the rostral confluence of the sinuses in mice, which we identified as a “hot spot” of iRBC accumulation / sequestration, has quite recently gained substantial interest. Functionally it bears similarities to perisinus spaces in humans. We shortly discuss these similarities in the MS. These may point – certainly speculative at the moment – to pathologies in the venous lacunae where iRBCs may sequester or passively accumulate with severe effects on efflux through the bridging veins. We discuss this shortly in the MS (page 19, line 442-453)

Q4: Can the authors suggest a mechanism by which accumulations of iRBCs, in the absence of an interaction with endothelial cells, would result in an inflammatory response? Could there be a rheologic effect (e.g., Herricks T, et al., Cell Microbiol 2012)? Could it be hydrostatic?

Response: In fact, we think that rheologic effects, alone or in combination with other anomalies, offer a likely explanation for the reduced flow, but less so for inflammation responses, which likely develop independently. It could be, as the recent Rodriguez-Munoz, D., *et al.* ref. 37 paper suggests, that inflammation and edema development are, at least partially, developing independently of each other. We have now included this in the discussion (page 16, line 381-391)

Q4: In human CM, there is abundant evidence of iRBCs in the cerebral microcirculation. Venous vessels which would normally empty after death are often distended by iRBCs. This is consistent with the authors’ observations of increased brain volume in association with the accumulations of iRBCs. It is unfortunate that the methods used in this manuscript were not sensitive enough to observe individual vessels. Absent that, it is difficult to determine if the decreased venous efflux originates in smaller, intracerebral vessels and the ensuing decreased flow results in accumulations of iRBCs in larger veins or if the accumulations of iRBCs in the larger draining veins and sinuses are the inciting events, and, in effect, ‘dam up’ flow from the upstream veins and post-capillary venules. Can the authors illuminate this ‘chicken and egg’ question? Which came first, obstruction of small vessels or the accumulation of iRBCs in larger draining veins and sinuses?

Response: We think that the reduction in flow in the large veins has a leading role, because this could, in our view, best explain the edema dynamics, the speed at which the edema develops and the affection of venous watershed areas. If obstructions start at a small vessel level one would

have to assume complex spatiotemporal cascades of small vessel affections, for which there seem to be, at least currently, no convincing explanations or precedence from other CNS pathologies. If one assumes rheopathological alterations, *e.g.* more viscous flows in the sinuses, this could rapidly result in severely increased intracranial pressure. In such a scenario, collateral flow will be fundamentally impaired, in contrast to local thrombi. It seems highly likely that pathologies related to such alterations in fluid dynamics are characterized by thresholds of the crucial parameters, up to which the flow can be normal, while a rapid switch to pathological states can occur once the thresholds are reached.

The temporal relationship between the administration of pyrimethamine and the abrogation of the inflammatory response is striking and is one of the real strengths of the paper.

Q5: What was the rationale for choosing pyrimethamine? It is an older antimalarial, and is rarely, if ever, used now. There are newer, faster drugs. The choice of this antimalarial should be justified by the authors.

Response: Pyrimethamine is the standard treatment in murine malaria models and routinely used to clear infections and select recombinant parasites, since all wild-type laboratory strains are highly sensitive to pyrimethamine. Combining a range of drugs with different clearance rates with the imaging techniques will clearly be a rewarding avenue for future studies.

Overall, this manuscript, by highlighting the potential importance of impaired venous efflux and the reversibility of the early changes associated with it, opens up important new areas of study related to CM pathogenesis and treatment. Establishing the temporal association between impaired venous drainage and subsequent pathophysiologically significant events (at least in the murine model) really changes the landscape and helps to explain earlier observations.

Terrie E. Taylor, D.O.

We thank the reviewer for the encouraging comments.

Reviewers' Comments:

Reviewer #2:

Remarks to the Author:

The authors have done a commendable job of addressing all my previous comments. The manuscript now presents an extremely detailed and thorough analysis of the larger vessels in ECM, which reveals multiple new insights into the vascular pathogenesis.

I have just a few residual comments:

Thank you for evaluating the changes in sequestration in the large draining veins in pbNK65-infected and perforin deficient C57Bl/6 mice and in BALB/c mice. I appreciate that this was a lot of extra work but it has substantially increased the value of the findings in the manuscript. Please can you show the course of parasitemia in the infections in C57Bl/6 mice and in BALB/c mice, to confirm that it is similar to that in the wild type C57Bl/6 mice? This is a pre-requisite for drawing conclusions about the impact of genetic background on the vascular pathogenesis. Assuming that the course of parasitaemia similar, it would be nice if the authors can speculate about how perforin may play a role in the vascular sequestration at D5, when there are already differences in the accumulation of infected red cells in the large draining veins, but not yet any evidence of "immunopathology".

Lines 276-277 "...hence, iRBC accumulation is a prerequisite for triggering ECM." This is a little ambiguous, but I read "accumulation" to mean the specific accumulation in the draining veins rather than a general increase in parasitemia. If this is the intended meaning, I do not agree with this conclusion. The experiments with pyrimethamine treatment show that ongoing infection (parasitemia) beyond D5 is essential for the development of ECM, and that the iRBC accumulation in draining veins at D5 is not sufficient to cause ECM if anti-parasite treatment is given at D5. It is not possible to conclude from this that iRBC accumulation in the large draining veins is the cause of ECM. To determine whether the accumulation (sequestration) of iRBCs is necessary for ECM it would be necessary to allow infection to proceed, but somehow "de-sequester" the iRBCs in the draining veins and restore normal blood flow between days 5-7. I am not suggesting that the authors need to do this, but they need to tone down the statement in lines 276-277 to recognise the current experimental approaches cannot definitively establish causation.

Panels labelling in Figure 1 should be reordered in the order they are mentioned in the text.

Figure 4 Legend. Line 517. The wild type images are on the left and the perforin-/- on the right?

Thanks for the opportunity to review this fascinating study, which certainly advances understanding of ECM and probably has relevance to understanding human CM.

Reviewer #4:

Remarks to the Author:

This revised manuscript has been highly responsive to the initial reviewer comments. The authors have carried out a number of additional experiments resulting in important new histological data, more in vitro data, findings from perforin-deficient mice (highlighting the contribution of immunecells) and from control mice (infected with malaria, but not susceptible to cerebral malaria).

As a result, they are able to further substantiate their primary findings regarding the importance of impaired venous efflux. They've also documented new details of venous drainage in the mouse model (two parallel venous streams) and have described the presence of infected red cells in the bone marrow of the skull.

This is important work and will help shape the direction of future research into the pathophysiology of CM.

All of my concerns were more than adequately addressed.

Response to reviewer comments: We thank the reviews for critically reading the manuscript and providing constructive feedback. We have addressed all the points raised by the reviewers by including the requested data and modifying the text according to the suggestions.

Reviewer #2:

Thank you for evaluating the changes in sequestration in the large draining veins in pbNK65-infected and perforin deficient C57Bl/6 mice and in BALB/c mice. I appreciate that this was a lot of extra work but it has substantially increased the value of the findings in the manuscript. Please can you show the course of parasitemia in the infections in C57Bl/6 mice and in BALB/c mice, to confirm that it is similar to that in the wild type C57Bl/6 mice? This is a pre-requisite for drawing conclusions about the impact of genetic background on the vascular pathogenesis.

Response: We thank the reviewer for the appraisal of our study. The course of parasitemia was comparable between the infected C57Bl/6 wt mice and BALB/c mice, indicating that the genetic background of the host determines the outcome of the vascular pathogenesis in ECM. The data are now included as new Fig. S2 B and discussed on page & lines 124-126.

Assuming that the course of parasitaemia similar, it would be nice if the authors can speculate about how perforin may play a role in the vascular sequestration at D5, when there are already differences in the accumulation of infected red cells in the large draining veins, but not yet any evidence of "immunopathology".

Response: We thank the reviewer for the suggestion we speculate that perforin may lead to endothelial activation, which may further foster accumulation of infected RBCs to the brain microvasculature. This is now discussed on page 14 lines 323 to 325.

Lines 276-277 "...hence, iRBC accumulation is a prerequisite for triggering ECM." This is a little ambiguous, but I read "accumulation" to mean the specific accumulation in the draining veins rather than a general increase in parasitemia. If this is the intended meaning, I do not agree with this conclusion. The experiments with pyrimethamine treatment show that ongoing infection (parasitemia) beyond D5 is essential for the development of ECM, and that the iRBC accumulation in draining veins at D5 is not sufficient to cause ECM if anti-parasite treatment is given at D5. It is not possible to conclude from this that iRBC accumulation in the large draining veins is the cause of ECM. To determine whether the accumulation (sequestration) of iRBCs is necessary for ECM it would be necessary to allow infection to proceed, but somehow "de-sequester" the iRBCs in the draining veins and restore normal blood flow between days 5-7. I am not suggesting that the authors need to do this, but they need to tone down the statement in lines 276-277 to recognise the current experimental approaches cannot definitively establish causation.

Response: We agree with the reviewer. we have toned down the statement that iRBC accumulation is a prerequisite for triggering experimental cerebral malaria and rephrase that sequestration of infected red blood cells and reduced venous efflux precede inflammation in experimental cerebral malaria (Page 12, lines 282-284).

Panels labelling in Figure 1 should be reordered in the order they are mentioned in the text.

Response: The panel labelling has been reordered in the order they are mentioned.

Figure 4 Legend. Line 517. The wild type images are on the left and the perforin-/- on the right?

Response: We thank the reviewer for pointing this out. The legend has been corrected.

Reviewer #4

This is important work and will help shape the direction of future research into the pathophysiology of CM. All of my concerns were more than adequately addressed.

Response: We thank the reviewer for the appraisal of our study.